

# Benefit of ozone observations from Sentinel-5P and future Sentinel-4 missions on tropospheric composition

Samuel Quesada-Ruiz[1,2,3], Jean-Luc Attié[1,2], William A. Lahoz[4], Rachid Abida[1], Philippe Ricaud[1], Laaziz El Amraoui[1], Régina Zbinden[1], Andrea Piacentini[5], Mathieu Joly[1], Henk Eskes[6], Arjo Segers[7], Lyana Curier[7], Johan de Haan[6], Jukka Kujanpää[8], Albert Oude-Nijhuis[6], Johanna Tamminen[8], Renske Timmermans[7], and Pepijn Veefkind[6]

[1]CNRM, Météo-France/CNRS UMR 3589, Toulouse, France
[2]Université de Toulouse, Laboratoire d'Aérologie, CNRS UMR 5560, Toulouse, France
[3]Currently at: European Centre for Medium-Range Weather Forecasts, Shinfield Park, Reading, RG2 9AX, UK
[4]NILU - Norwegian Institute for Air Research, P.O. Box 100, 2027 Kjeller, Norway
[5]CERFACS, Global Change and Climate Modelling Team, 31057 Toulouse, France
[6]Royal Netherlands Meteorological Institute (KNMI), P.O. Box 201, 3730 AE De Bilt, The Netherlands
[7]TNO, Business unit Environment, Health and Safety, P.O. Box 80015, 3508 TA Utrecht, The Netherlands
[8]Finnish Meteorological Institute, Earth Observation Unit, P.O. Box 503, 00101 Helsinki, Finland

**Correspondence:** Jean-Luc Attié (jean-luc.attie@aero.obs-mip.fr)

**Abstract.** We present an observing simulated system experiment (OSSE) dedicated to evaluate the potential added value from the Sentinel-4 and the Sentinel-5P observations on tropospheric ozone composition. For this purpose, the ozone data of Sentinel-4 (Ultraviolet Visible Near-infrared) and Sentinel-5P (TROPOspheric Monitoring Instrument) onboard a geostationary (GEO) and a low Earth orbit (LEO) platform, respectively, has been simulated for the summer 2003. To ensure the robustness of

the results, the OSSE has been configured with conservative assumptions. We simulate the reality by combining two chemistry transport models (CTMs): the Long Term Ozone Simulation-European Operational Smog (LOTOS-EUROS) and the Transport Model version 5 (TM5). The assimilation system is based on a different CTM, the MOdèle de Chimie Atmosphérique à Grande Echelle (MOCAGE), combined with the 3D variational technique. The background error covariance matrix does not evolve in time and its variance is proportional to the field values. The simulated data are formed of six eigenvectors to minimize the

size of the dataset by removing the noise-dominated part of the observations. The results show that the satellite data clearly bring direct added value around 200 hPa for the whole assimilation period and for the whole European domain, while a likely indirect added value is identified but not for the whole period and domain at 500 hPa, and to a lower extent at 700 hPa. In addition, the ozone added value from Sentinel-5P (LEO) appears close to that from Sentinel-4 (GEO) in the free troposphere (200 - 500 hPa) in our OSSE.

# 1   Introduction

The monitoring of tropospheric composition is of utmost importance for the evaluation of air quality and to improve the understanding of the intercontinental transport of air pollution (HTAP, 2010). Recently, satellite measurements have been widely used to improve the detection and the forecast of atmospheric pollutants through their assimilation into chemistry transport



models (CTMs), including ground-based and/or airborne measurements (e.g., Elbern et al., 2010). The main advantage of satellite observations, when compared to local measurements, is the global and/or regional coverage. However, the temporal and spatial resolution required for air quality (AQ) is still a drawback that should be addressed by future missions in order to reach up to 10 km resolution and up to 1 hour of revisit time. To address these issues, studies have analysed the combined use

of various geostationary Earth orbit (GEO) / low Earth orbit (LEO) satellites (e.g., Lahoz et al., 2012; Barré et al., 2015).

Copernicus, the European programme for the establishment of a European capacity for Earth observation (http://www.copernicus.eu/), largely relies on data from satellites observing the Earth. Particularly, the Sentinel-4 (S4), the Sentinel-5 (S5) and Sentinel-5 Precursor (S5P) missions are dedicated to monitoring the atmospheric composition for Copernicus Atmosphere Services -CAMS- (http://www.esa.int/Our_Activities/Observing_the_Earth/Copernicus/Sentinels_-4_-5_and_-5P).

The S4 mission will be carried on the Meteosat Third Generation (MTG) geostationary platform, and includes an ultraviolet visible near-infrared (UVN) spectrometer. The S5P mission, launched on 13 October 2017, includes the Tropospheric Monitoring Instrument (TROPOMI) and was developed to reduce the gap between the Scanning Imaging Absorption Spectrometer for Atmospheric Cartography (SCIAMACHY) instrument on Envisat, the Ozone Monitoring Instrument (OMI) on Aura mission and the Sentinel-5 (S5) mission. The work presented here was part of a project called Impact of Spaceborne Observations on

Tropospheric Composition Analysis and Forecast (ISOTROP, http://projects.knmi.nl/isotrop/), financed by the European Space Agency (ESA) to study the impact of trace gas observations by the Sentinel missions on air quality analyses.

The analysis of the benefit of a trace gas observation -in our case ozone- is carried out by performing an observing system simulation experiment (OSSE). The main goal of the OSSE concept is to determine the potential added value of a new observing system (OS) with respect to the existing ones. We use a state-of-the-art model (namely a CTM) run to construct a representation

of reality, called hereafter the nature run (NR). From the NR, the satellite trace gas observations (level 2 data) and their corresponding errors are simulated using an instrument simulator, which combines a retrieval scheme and the instrument model. These simulated observations are preferably fed into a data assimilation system of a different model, obtaining an assimilation run (AR). This second model is also run either without assimilation or with the assimilation of the existing OS data, which gives a reference run (RR). The use of two different models avoids the identical twin problem, which is known to

lead to overoptimistic results (Arnold Jr and Dey, 1986; Timmermans et al., 2015). Numerous OSSEs dealing with observations of chemical species were performed using different satellite instrument specifications to show the benefit of selected additional observations on the OS (e.g., Edwards et al., 2009; Claeyman et al., 2011; Zoogman et al., 2011; Abida et al., 2017).

In this study, we perform an OSSE to analyse the benefit of tropospheric ozone observations from S4 and S5P missions, following the recommendations for an AQ OSSE reported by Timmermans et al. (2015). We used as level 2 observations

the nadir simulated ozone measurements in the ultraviolet (UV) from the future S4 and the current S5P missions. It is worth pointing out that this work started before launch of S5P, and its value includes the comparison with S4. To be consistent with S4 and the studied period, we simulated S5P ozone data and we used the same UV spectral range for both. Note that the results presented in this study correspond to the use of instrumental characteristics of a S4-like and S5P-like missions that are assumed to be consistent with the actual characteristics of S4 and S5P missions, but for the sake of simplicity we will call them

hereinafter S4 and S5P. In addition, we also simulated ground-based stations (GBS) ozone data to evaluate the added value





of the satellite measurements within the lower troposphere in comparison to ground-based data. The ozone data are simulated from the NR, which is formed by the combination of the CTMs Long Term Ozone Simulation-European Operational Smog -LOTOS-EUROS- (Manders et al., 2017) and Transport Model version 5 -TM5- (Huijnen et al., 2010). These ozone data were assimilated in the MOdèle de Chimie Atmosphérique à Grande Echelle-Projet d'Assimilation par Logiciel Multi-méthodes

(MOCAGE-PALM) system (Peuch et al., 1999; Lagarde et al., 2001) to provide both the ARs and the RR that are compared to the NR. In our case, the RR is the assimilation of the GBS simulated data. The ARs include the assimilation of GBS and satellite simulated data (S4 or S5P or both S4 and S5P - we note hereafter S4+S5P).

For this OSSE, we selected the summer 2003 (June, July and August). During this period, a heat wave affected Europe, especially during the first two weeks of August, leading to the hottest summer recorded since the 16th century (Stott et al., 2004;

García-Herrera et al., 2010). Note that our study does not take into account the heat wave of 2018, whose impact has not been fully assessed yet. Various studies suggested that from 40,000 to 70,000 deaths during this heat wave were attributed to heat and pollution in Europe (Robine et al., 2008; García-Herrera et al., 2010). The heat wave caused elevated ozone concentrations, due to its close correlation with high temperatures, that were enhanced by the anticyclonic conditions. High temperatures and clear sky conditions during summer time are advantageous conditions for ozone precursor photochemical reactions, especially

over populated areas, where anthropogenic ozone precursor emissions (nitrogen oxides -NOx- or carbon monoxide -CO-) are predominant. Regarding the heat wave consequences related to AQ, surface ozone measurements over central Europe were the highest since 1980s (Solberg et al., 2008). Additionally, unprecedented forest fires in Portugal occurred, emitting huge quantities of CO (Abida et al., 2017). High ozone and CO concentrations were also measured by MOZAIC (Measurements of OZone, water vapour, carbon monoxide and nitrogen oxides by Airbus In-service airCraft) instruments on board commercial

aircraft as reported by Tressol et al. (2008).

The general aim of this paper is to assess the benefit of future ozone data from individual or combined use of GEO (S4) and LEO (S5P) satellite observations for the understanding of local to regional scale ozone tropospheric composition with a focus on Europe. Section 2 describes the MOCAGE-PALM assimilation system. We define the OSSE components, including the NR, the RR and the AR, in Sect. 3. We present the simulated ozone measurements in Sect. 4 and the metrics used to evaluate our

OSSE in Sect. 5. We show the results of the ozone OSSE at different altitudes from the upper to the lower troposphere for the summer 2003 period in Sect. 6 which are discussed in Sect. 7, before concluding in Sect. 8.

## 2   The assimilation system

The assimilation system used in this study, MOCAGE-PALM, was jointly developed by Météo-France and the Centre Européen de Recherche et de Formation Avancée en Calcul Scientifique (CERFACS). The MOCAGE-PALM assimilation system has

been used in several studies related to upper tropospheric and stratospheric ozone (El Amraoui et al., 2008a, b), in ozone OSSEs related to air quality (Claeyman et al., 2011) and to evaluate the quality of IASI (Infrared Atmospheric Sounding Interferometer) total column ozone measurements (Massart et al., 2009).



MOCAGE (Peuch et al., 1999) is a 3D CTM that reproduces the main chemical and physical processes present in the troposphere and the stratosphere (Bousserez et al., 2007). From the various configurations, domains, grid resolutions and chemical/physical parametrizations available in MOCAGE, the following were selected: (i) a 2°x2° spatial resolution global grid using a 2-way nesting with a 0.2°x0.2° spatial resolution regional grid (32-72° N, 16° W - 36° E), (ii) 47 sigma-hybrid

vertical levels from the surface up to 5 hPa, and (iii) the RACMOBUS chemical scheme. The RACMOBUS chemical scheme is the combination of the Regional Atmospheric Chemistry Mechanism tropospheric scheme -RACM- (Stockwell et al., 1997) and the REactive Processes Ruling the Ozone BUdget in the Stratosphere stratospheric scheme -REPROBUS- (Lefèvre et al., 1994). MOCAGE is used for diverse purposes: e.g., operational chemical weather forecasts (Dufour et al., 2005), Monitoring Atmospheric Composition and Climate (MACC) services (http://www.gmes-atmosphere.eu), and atmospheric composition cli-

mate trends studies (Teyssèdre et al., 2007). Moreover, during the heat wave of August 2003, MOZAIC aircraft measurements were used to validate MOCAGE ozone fields over Europe (Ordóñez et al., 2010).

The data assimilation suite is run via the PALM coupler (Lagarde et al., 2001) that connects the CTM to a set of operators, such as the observation operators, the error covariance approximations, the increment propagators, and the minimizer, implementing several variational assimilation algorithms. We used the 3D-Var variant, which has already been presented for a S5P

CO OSSE (Abida et al., 2017), with a fixed assimilation window of one hour.

The 3D spatial distribution of the information in the data is determined by the background error covariance matrix (B-matrix), that depends only on the model and has a large impact on the analysis. In our assimilation system, the B-matrix is based on the diffusion equation (Weaver and Courtier, 2001). We estimated the B-matrix by means of a simple parametrization, using a 3D variance field and 3D fields of the horizontal and vertical local correlation length scales. Our B-matrix does not

evolve in time and its variance is related to the background values. The variance is calculated as the square of 25% of the background profile (from the surface to the top of the atmosphere). Concerning the off diagonal terms of the B-matrix, we defined two horizontal local correlation lengths considered as constant and equal to 0.4° (two model grid boxes), and a vertical correlation length set to one vertical model grid level. The goal of these lengths is to spread the information in the data on the horizontal and the vertical.

## 25   3   Description of the OSSE components

We present a regional chemical OSSE framework conducted to investigate the added value of S4 and S5P observations on tropospheric ozone. In addition to simulated satellite observations, the ground based station (GBS) ozone data are assimilated into our system to reproduce the existing OS.We describe the OSSE scheme in Fig. 1 with the links between the different elements: the observations, the free run (FR), the nature run (NR), the reference run (RR) and the assimilation run (AR). The

added value of the simulated observations is evaluated through comparison of the AR and the RR with respect to the NR. This experiment was aimed to be as less overoptimistic as possible. For this reason, the RR and the AR (see Sect. 3.2) are generated from another model than the NR model(s) to avoid the identical twin problem.





### 3.1 The nature run

The selection of the nature run OSSE component (Fig. 1) is of utmost importance. The NR model characterizes the true state of the atmospheric composition. A CTM is often used to simulate the NR (Masutani et al., 2010) and, in turn, the NR is used to simulate the reference state through a data simulator that includes the retrieval method and the instrument model. In this

study, the ozone NR is made up of two different models. On the one hand, the global CTM TM5 (Huijnen et al., 2010) is run over Europe with a spatial resolution of 1°x1°, a temporal output resolution of 3 hours, and 34 vertical layers from the surface up to 0.1 hPa. On the other hand, the regional LOTOS-EUROS AQ model (Manders et al., 2017) provides a description of the lowermost tropospheric air pollution over Europe, with a 7-km spatial resolution, a 1-hour temporal output resolution, and 4 vertical layers from the surface up to 3.5 km. The European Centre for Medium-Range Weather Forecasts (ECMWF)

meteorological data are used as input for both LOTOS-EUROS and TM5. The MACC global fire assimilation system (GFAS v1, Kaiser et al., 2012) is used for fire emissions in both models, and the TNO-MACC-II emission database (Kuenen et al., 2014) for surface anthropogenic emissions in LOTOS-EUROS. The model runs include a spin-up period of three months. The NR is built by merging the LOTOS-EUROS ozone profiles from the surface to 3.5 km with the TM5 results from 3.5 km to the top of the model atmosphere.

The ozone representation within TM5 and LOTOS-EUROS has been validated in Van Loon et al. (2007) for the year 2001. In addition, TM5 ozone data were evaluated against MOZAIC aircraft measurements during the heat wave of August 2003 by Ordóñez et al. (2010). For LOTOS-EUROS validation, we used real data from a randomly selected subset of GBS over Europe based on the following criteria: i) sites qualified as "rural background", according to the metadata, that fall into the first five classes (over ten) of the objective classification using the method from Joly and Peuch (2012), and ii) stations with at

least 90% of the hourly data in each month in the studied time period. Figure 2 shows the performance for LOTOS-EUROS model with respect to the surface ozone concentrations over 325 stations from the European air quality database (AirBase) dataset during August 2003. Two distinct periods of time in August 2003 can be identified, one from 1st to 14th of August characterized by elevated surface ozone concentration showing a ozone background enhancement of ∼40 µg.m$^{-3}$ compared to the second period. The surface ozone concentrations from AirBase (black dots) illustrate the greater amplitude of the daily

variation in the first period. Clearly, the LOTOS-EUROS model (blue lines) has a larger 2-week amplitude compared to the second period. The second period exhibits more typical ozone concentrations well represented by LOTOS-EUROS. The bias between LOTOS-EUROS and the surface ozone measurements is about 10 to 20 µg.m$^{-3}$. However, the general behaviour of the LOTOS-EUROS diurnal cycle is similar to that of the GBS measurements.

### 3.2 The reference run and the assimilation run

The reference run (RR) -another essential component in the design of an OSSE- includes the assimilation of the existing OS data. In this OSSE, the GBS represent the existing OS at the surface. Therefore, in order to account for the impact of the existing OS, we assimilated the simulated ozone observations from GBS using MOCAGE-PALM as it is done operationally. In addition, as stated previously, a well designed OSSE should use a different model to generate the RR than the one used for





the NR. In our work, we generated the RR from the MOCAGE model constrained by the meteorological data from the Action de Recherche Petite Echelle Grande Echelle (ARPEGE) model (Courtier et al., 1991), which is different from the two models used to construct the NR.

Concerning the assimilation run (AR), we assimilated the simulated satellite (S4, S5P) ozone data and GBS measurements
derived from the NR using the assimilation system MOCAGE-PALM. Three assimilation runs (ARs) were performed:

i) the S4 ozone AR (hereafter called S4_AR) which is the simultaneous assimilation of simulated ozone from S4 and GBS, to evaluate the added value of S4 ozone with respect to the existing OS ;

ii) the S5P ozone AR (hereafter called S5P_AR) which is the simultaneous assimilation of simulated ozone from S5P and GBS, to evaluate the added value of S5P ozone with respect to the existing OS;

iii) the S4 and S5P ozone AR (hereafter called S4+S5P_AR) which is the simultaneous assimilation of simulated ozone from S4, S5P and GBS, to evaluate the synergy of the combined use of S4 and S5P ozone.

## 4  Description of the simulated ozone data

In this section, we discuss how we simulated the in situ and the satellite data from the NR.

### 4.1  The simulated GBS ozone data

We derived the ground-based simulated ozone data from the surface representation of the NR (LOTOS-EUROS model). The locations of the stations are taken from the AirBase dataset. The GBS data at these locations are routinely used in the operational system to forecast AQ (e.g. MACC reanalysis). The GBS are sorted out by keeping the stations representative of the background ozone (urban and rural). We used 1132 stations measuring ozone over Europe (Fig. 3) to preserve an homogeneous spatial representativity. The observation error is taken to be 5 parts per billion by volume (ppbv) in our assimilation system, in
agreement with Jaumouillé et al. (2012).

### 4.2  The simulated S4 and S5P ozone data

Synthetic satellite ozone profile observations were generated from the NR based on S5P instrument model and characteristics for both S4 and S5P. The equatorial overpass time of S5P (13:30 local time of ascending node crossing) was adopted for the LEO simulations while the GEO data were generated using the S4 measurement geometry and an hourly measurement revisit
time.

We focus on tropospheric ozone, and in our study the retrievals were performed in the 300-320 nm spectral window. We assumed a spectral resolution Full Width at Half Maximum (FWHM) of 0.5 nm, a signal-to-noise ratio (SNR) of 5000 for the solar irradiance measurement, and linear interpolation between the following SNR values for the Earth radiance measurement: 50, 300 and 1000 at 300, 310 and 320 nm, respectively. The selected spectral window allows retrievals at the full spatial
sampling of about 7x7 km$^2$ using the S4 UV-VIS or the TROPOMI UV2 channel. We note that the spatial sampling of





TROPOMI has been increased to 3.5x7 km$^2$ before launch. The TROPOMI UV1 channel measuring at 270-300 nm is sensitive to the stratosphere but has much larger ground pixels of about 21x28 km$^2$, and is not included in the S4 characterization. The ozone degrees of freedom for signal (DFS) is of the order of 4-5 for this wavelength range for both S4 and S5P. Removing the lower wavelength radiance measurements (270-300 nm) from the retrieval has only a small impact on the DFS in the troposphere, but reduces the DFS in the middle/higher stratosphere.

The Determining Instrument Specifications and Analyzing Methods for Atmospheric Retrieval (DISAMAR) package (de Haan, 2015) is used for ozone profile retrievals involving forward model radiative transfer calculations to simulate the measured spectrum followed by the optimal estimation (OE) method to retrieve the profiles (Rodgers, 2000). The retrieval results are stored in a compact form following the approach outlined by Migliorini et al. (2008). The basic idea of Migliorini et al. is a representation of the optimal estimation retrieval results in a way which greatly reduces the volume of the data product without losing the information, and which leads to an efficient interface with data assimilation by reducing the number of observations (removing the noisy part of the retrieval solution). This is done by the following steps:

  i) Transforming the retrieval state in such a way that the a-priori does not explicitly enter the observation operator.

  ii) The retrieval solution is expressed in the space of the eigenvectors of the retrieval problem.

  iii) A rescaling of the final eigenvectors such that the noise becomes equal to 1.0.

  iv) A removal of noise-dominated observations with no information.

As a result the a-priori vector and the covariance matrix (unity matrix) do not have to be stored, and only a truncated generalised averaging kernel is written to file, together with the retrieved values for the dominant states.

The transformations of the Migliorini et al. approach have specific aspects depending on the way the optimal estimation is implemented. Therefore we will provide details on how this was implemented for our ozone case. In DISAMAR, a basis transformation known as pre-whitening is applied leading to a transformed Jacobian $\overline{\mathbf{K}}$

$$\overline{\mathbf{K}} = \mathbf{S}_\varepsilon^{-1/2} \mathbf{K} \mathbf{S}_a^{1/2} = \mathbf{U} \mathbf{W} \mathbf{V}^T \tag{1}$$

where $\mathbf{S}_\varepsilon$ and $\mathbf{S}_a$ are the measurement and a-priori covariance matrices, respectively, and $\mathbf{K}$ the Jacobian matrix. On the right side, the singular value decomposition (SVD) has been applied. The diagonal matrix $\mathbf{W}$ contains the singular values $w_k$ while the matrix $\mathbf{V}$ contains the singular vectors ($\mathbf{U}$ is not used here). The retrieved profile $\mathbf{x}$, a-priori profile $\mathbf{x}_a$ and the true profile $\mathbf{x}_{true}$ are connected by the averaging kernel $\mathbf{A}$

$$\mathbf{x} - \mathbf{x}_a = \mathbf{A}\left(\mathbf{x}_{true} - \mathbf{x}_a\right) + \mathbf{G}\varepsilon, \tag{2}$$

where $\mathbf{G}$ is the gain matrix and $\varepsilon$ describes the measurement noise. The covariance of the noise $\mathbf{S}_{noise}$ is given by

$$\mathbf{S}_{noise} = \mathbf{G}\mathbf{S}_\varepsilon\mathbf{G}^T = \mathbf{S}_a^{1/2}\mathbf{V}(\mathbf{G}')^2\mathbf{V}^T\mathbf{S}_a^{1/2}, \tag{3}$$



where the transformed gain matrix $\mathbf{G}'$ is a diagonal matrix formed of the singular values $w_k$

$$\mathbf{G}' = \mathrm{diag}\left\{\frac{w_k}{w_k^2 + 1}\right\} \tag{4}$$

The retrieval solution $\mathbf{x}$ is transformed three times. The first transformation shifts the solution removing the need to provide the a-priori profile

$$\mathbf{x}^{(a)} = \mathbf{x} - [\mathbf{I} - \mathbf{A}]\mathbf{x}_a = \mathbf{A}\mathbf{x}_{true} + \mathbf{G}\varepsilon, \tag{5}$$

where $\mathbf{x}^{(a)}$ is the shifted solution having the same $\mathbf{S}_{noise}$ as $\mathbf{x}$. The second transformation rotates the shifted solution

$$\mathbf{x}^{(b)} = \mathbf{V}^T\mathbf{S}_a^{-1/2}\mathbf{x}^{(a)} = \mathbf{V}^T\mathbf{S}_a^{-1/2}\mathbf{A}\mathbf{x}_{true} + \mathbf{V}^T\mathbf{S}_a^{-1/2}\mathbf{G}\varepsilon \tag{6}$$

to obtain a diagonal covariance $\Lambda$

$$\Lambda = \mathbf{V}^T\mathbf{S}_a^{-1/2}\mathbf{S}_{noise}\mathbf{S}_a^{-1/2}\mathbf{V} = \mathrm{diag}\left\{\left[\frac{w_k}{w_k^2 + 1}\right]^2\right\}. \tag{7}$$

The storage of the covariance matrix can be avoided by scaling the rotated solution $\mathbf{x}^{(b)}$

$$\mathbf{x}^{(c)} = \Lambda^{-1/2}\mathbf{x}^{(b)} = \Lambda^{-1/2}\mathbf{V}^T\mathbf{S}_a^{-1/2}\mathbf{A}\mathbf{x}_{true} + \varepsilon^{(c)} = \mathbf{A}^{(c)}\mathbf{x}_{true} + \varepsilon^{(c)} \tag{8}$$

where the covariance of $\varepsilon^{(c)}$ is the identity matrix $\mathbf{I}$. The transformed solution $\mathbf{x}^{(c)}$ is obtained from the retrieval parameters using

$$\mathbf{x}^{(c)} = \Lambda^{-1/2}\mathbf{V}^T\mathbf{S}_a^{-1/2}\left(\mathbf{x} - [\mathbf{I} - \mathbf{A}]\mathbf{x}_a\right) \tag{9}$$

while the transformed averaging kernel $\mathbf{A}^{(c)}$ is obtained from

$$\mathbf{A}^{(c)} = \Lambda^{-1/2}\mathbf{V}^T\mathbf{S}_a^{-1/2}\mathbf{A} = \Lambda^{-1/2}\mathbf{G}'\mathbf{W}\mathbf{V}^T\mathbf{S}_a^{-1/2} \qquad = \mathrm{diag}\{w_k\}\mathbf{V}^T\mathbf{S}_a^{-1/2}. \tag{10}$$

To further reduce storage space requirements, only the leading $q$ eigenvectors are provided by storing only the first $q$ elements of $\mathbf{x}^{(c)}$ and the first $q$ rows of $\mathbf{A}^{(c)}$.

In our case, we considered the first six leading eigenvectors ($q = 6$), hereafter labelled as v1 to v6. Figure 4 presents the average over the European domain for the summer 2003 of the S4 $\mathbf{A}^{(c)}$ (the S5P one is similar but not shown). The averaging kernel (AK) representative of the first eigenvector (v1), i.e. the first row of $\mathbf{A}^{(c)}$, exhibits a broad maximum from the surface up to 1 hPa peaking at about 40 hPa. This AK is positive everywhere, and is representative of the ozone column amount in the lower stratosphere. The AK representative of the second eigenvector (v2) exhibits a maximum at about 6 hPa, distinguishing middle-stratosphere ozone from lower stratosphere ozone. The AKs of the other eigenvectors are much smaller but these include crucial tropospheric information. As can be seen from the right panel, they provide information above the altitude of 10 hPa (v3) and, more importantly, also below 100 hPa (v3 and v4). The AKs for v5 and v6 show small values for all the levels and the absolute value becomes small compared to the noise level (=1). Comparing the set of AKs, the tropospheric





information is likely contained in v1, v3 and v4, while the highest sensitivity of the retrieval is in the stratosphere. However, we used the first six leading eigenvectors (v1-v6) to keep nearly all the tropospheric information contained in the eigenvectors. Moreover, keeping the v1 to v6 safely represents the DFS which is typically of the order of 4-5.

Performing the OE retrieval for each measurement requires excessively large computational resources. For our simulation study, we simplified the retrieval process by introducing look-up tables (LUTs) for $\mathbf{A}^{(c)}$ and $\mathbf{x}^{(c)}$. Using the U.S. standard atmosphere temperature and ozone profiles, $\mathbf{A}^{(c)}$ was obtained for LUT nodes for solar zenith angle (sza), viewing zenith angle (vza), relative azimuth angle, cloud or surface pressure, and surface albedo. The LUT has 10 solar zenith angles (cos(sza) = 0.1, 0.2, 0.3, 0.4, 0.5, 0.6, 0.7, 0.8, 0.9, 1), 8 viewing zenith angles (cos(vza) = 0.3, 0.4, 0.5, 0.6, 0.7, 0.8, 0.9, 1), 4 relative azimuth angles ( 0, 60, 120, 180 ), 9 surface (cloud) albedo points (0.02, 0.04, 0.06, 0.1, 0.2, 0.3, 0.4, 0.8, 0.9), 10 surface (cloud) pressure points (1050, 970, 890, 801, 701, 601, 501, 401, 301, 201). Note that the eigenstates and kernel rows are determined up to a plus or minus sign. Jumps from one sign to the other from one LUT point to the next will give problematic interpolation errors. Therefore an extra post-processing was applied to the LUT by checking the sign of neighbouring points in the LUT and by multiplying the kernel vectors by -1 when needed.

Using the LUT, the synthetic satellite ozone observations are generated by:

i) Generating the orbit coordinates and individual pixel coordinates with an orbit simulator, for both S4 and S5P.

ii) Interpolating ECMWF high-resolution (cloud, temperature) meteorological fields to these orbits, and obtaining radiative cloud fractions and cloud heights.

iii) Interpolating the NR results to the observation locations to obtain $\mathbf{x}_{true}$.

iv) Interpolating $\mathbf{A}^{(c)}$ from the LUT using the measurement geometry, cloud/surface pressure and cloud/surface albedo, weighting the result by radiative cloud fraction.

v) Computing the observation from $\mathbf{x}_{obs} = \mathbf{A}^{(c)}\mathbf{x}_{true}+$ noise. The noise realisation is drawn from a Gaussian distribution with unit width.

### 4.2.1 Satellite observations error covariance matrix

The total observation error results from a sum of the observation error, as provided in the synthetic observations data product, and a representativeness error (Migliorini et al., 2008), that will be explained later in this section.

The synthetic observations are provided with an observation error $\varepsilon^{(c)}$ added, drawn randomly from a normal distribution with a covariance matrix equal to the identity matrix and a transformed AK, $\mathbf{A}^{(c)}$. The ozone retrievals are presented in the space of eigenvectors, and the data product contains the first six leading eigenvectors. The AKs are unique, and are computed for each observation separately, and depend on the satellite geometry, surface albedo and cloud properties. The absolute value of this retrieval (the eigenvectors can contain negative values) is roughly a measure of the SNR, since the observation error is equal to 1 -in ln-concentration space- by construction. Therefore the observation error covariance matrix is the identity matrix.



**Table 1.** Monthly observation standard deviation, in $\ln(\mathrm{vmr_{ppm}})$, for S4 and S5P. The observation error covariance matrix (**R**) used in our assimilation system for summer 2003 is a diagonal matrix in which the diagonal elements correspond to the variance (the square of the values present in this table).

| | | Eigenvectors | | | | | |
|---|---|---|---|---|---|---|---|
| **Month** | **Satellite** | v1 | v2 | v3 | v4 | v5 | v6 |
| June | S4 | 74.0 | 1.6 | 1.2 | 1.0 | 1.0 | 1.0 |
| | S5P | 72.5 | 1.2 | 1.1 | 1.0 | 1.0 | 1.0 |
| July | S4 | 74.0 | 1.6 | 1.3 | 1.0 | 1.0 | 1.0 |
| | S5P | 73.5 | 1.8 | 1.3 | 1.0 | 1.0 | 1.0 |
| August | S4 | 77.0 | 1.8 | 1.2 | 1.0 | 1.0 | 1.0 |
| | S5P | 74.0 | 2.0 | 1.1 | 1.0 | 1.0 | 1.0 |

The representativeness error is often difficult to estimate and could describe, for instance, the mismatch between the satellite footprint and the model grid box, and also the differences in the information content between the satellite and the model vertical layers. Furthermore, other assumptions and inaccuracies in the observation operator (which transforms the model state into the observation space) also contribute to the representativeness error. Ceccherini et al. (2018) showed the importance of interpolation and coincidence errors for retrievals on different vertical grids in data fusion. For example, the difference in the layers between the retrieval grid and the MOCAGE CTM may easily lead to regridding (interpolation) errors that may make it difficult to assimilate stratospheric (high concentration) and tropospheric ozone (low concentration) together.

We estimate the representativeness error by using the NR provided in the retrieval product. The representativeness contribution to the covariance matrix is computed by taking into account the NR profiles and by calculating the corresponding variances between the retrieved ozone leading eigenvectors for each satellite instrument (S4 or S5P), $x^{(c)}$, and the NR profile, $x_{true}$, on a monthly basis. This calculation is done in the transformed observation space by applying the transformed AK, $\mathbf{A}^{(c)}$, to the NR. The diagonal of the observation error covariance matrix (**R**) is calculated as:

$$\mathrm{diag}(\mathbf{R}) = \frac{1}{N} \sum \left( \mathbf{x}^{(c)} - \mathbf{A}^{(c)} \mathbf{H} \mathbf{x}_{true} \right)^2 \tag{11}$$

where **H** is a linear spatial interpolator and N is the total number of pixels over the European domain within each summertime month. The values of the corresponding monthly standard deviation obtained are presented in Table 1 for S4 and S5P. Because random errors were added to the synthetic observations, we average the data each month to have robust statistics and also to take into account the possible change from month to month (intraseasonal variability).

The positive impact of including the representativeness error in **R** on the analysed ozone data has been evaluated for the assimilation of S4 ozone for the month of June using the values from Table 1. Using these values reduces the ozone weight in the stratosphere favouring the ozone assimilation in the troposphere and allows a stable combined assimilation of the GBS





ozone observations together with the S4 and S5P satellite data and gives a stable normalized $\chi^2$ statistics for the assimilation, with values ranging between 0.6 and 0.7 (not shown). Note that the values for eigenvectors v4 to v6 are unchanged (equal to 1 ln concentration-space). The information contained in the first three leading eigenvectors (v1 to v3) have higher SNRs compared to the others (v4 to v6), leading to a larger absolute error. The higher the SNR, the larger the representativeness

error. Conversely, the relative error ($\frac{\mathrm{diag}(\mathbf{R})}{\frac{1}{N}\sum|\mathbf{A}^{(c)}\mathbf{H}\mathbf{x}_{true}|}$) remains constant with 1% for the six leading eigenvectors (v1 to v6).

Figure 5 shows the histograms of Observation minus Analysis (OmA) and Observation minus Forecast (OmF) for the first six leading eigenvectors using $\mathbf{R}$ in the assimilation process for S4 (S5P ones are similar but not shown) during the month of June 2003. One can see clearly that the OmA histograms are narrower than the OmF for the four first leading eigenvectors (v1-v4). This shows that these eigenvectors have more impact on our assimilation system than the two others likely due to the

information representing greater ozone concentration, in particular for the v1.

In agreement with the conclusions of Migliorini et al. (2008), this sensitivity study shows the need to add a representativeness error to the observation covariance matrix in order to improve the assimilation. In our case, this is especially noticeable for the first three eigenvectors that contain most of the ozone information.

## 5   Metrics

We calculate the mean bias error (MBE), the mean absolute error (MAE), the root mean square error (RMSE) and its reduction rate or skill score, and the correlation coefficient to quantify the bias, the error and the agreement between the NR and the RR, or between the NR and the AR. The statistical indicators MBE, MAE, RMSE, skill score and correlation coefficient with respect to the NR, are defined as follows:

$$MBE(X) = \frac{1}{N}\sum(X - X_{NR}) \tag{12}$$

$$MAE(X) = \frac{1}{N}\sum|X - X_{NR}| \tag{13}$$

$$RMSE(X) = \sqrt{\frac{1}{N}\sum(X - X_{NR})^2} \tag{14}$$

$$Skill(X) = 1 - \frac{RMSE(X)}{RMSE(X_{RR})} \tag{15}$$

$$Correlation(X) = \frac{\sum\left(X - \overline{X}\right)\left(X_{NR} - \overline{X_{NR}}\right)}{\sqrt{\sum\left(X - \overline{X}\right)^2\sum\left(X_{NR} - \overline{X_{NR}}\right)^2}} \tag{16}$$



where $X$ can be $X_{RR}$ or $X_{AR}$, representing the RR or the AR data, respectively; $X_{NR}$ represents the NR data; $N$ is the total number of data samples; and the over-bar symbol represents the arithmetic mean operator. The data selection for $X$ will depend on the chosen comparison.

The MBE gives the average value by which the RR, or the AR differs from the NR over the entire dataset. The MAE and the

RMSE provide a measure of the error between the RR and the NR or between the AR and the NR. The RMSE gives a greater weight to large errors than the MAE. The skill score represents the reduction rate of the RMSE of the AR with respect to the RMSE of the RR. Its value ranges from negative infinity to 1. Ideally, a skill score of 1 means that the AR is equal to the NR. A positive skill score indicates that the error of the AR is lower than the error of the RR when compared to the NR, suggesting an added value from the assimilated data. Conversely, the skill score is negative when the error of the AR is larger than that of the

RR, which means the assimilation of the data degrades the analysis. Finally, when the skill score is 0 there is no improvement of the AR compared to the RR.

We use the correlation coefficient of the time series for each model grid box over the studied domain to measure the linear dependence between the RR and the NR or between the AR and the NR. A benefit from the satellite observations is identified when the correlation coefficient of the AR is greater that of the RR. Furthermore, we calculate the histogram of the correlation

coefficient of the time series for the considered period. If the histogram is narrow and peaks to 1, the time series of the NR and the AR are highly correlated, otherwise the time series of each grid box are less correlated.

## 6    Results

We perform the OSSE for the summer 2003. Because of the large number of satellite data, we only keep more reliable clear sky pixels by discarding the cloudy pixels (cloud fraction greater than 0.05). Moreover, during the summer 2003, the sky was

often clear and the addition of cloudy pixels has a negligible impact in the results. In addition, we perform a data thinning in order to minimize the spatial correlation between the observation errors, keeping one satellite observation each four model grid boxes in both latitude and longitude directions. The same procedure was applied to both S4 and S5P data.

In this OSSE, we propose to assess the added value of S5P and S4 ozone with respect to the existing OS during the study period, i.e., the GBS network. The goal is to study the benefit of assimilating these future and/or current satellite data on

tropospheric ozone. For this purpose, we compare the three assimilation runs (S4_AR, S5P_AR, S4+S5P_AR) with respect to the RR and the NR.

Figure 6 shows the RR, the three ARs and the NR mean ozone profiles for the summer 2003, and for June, July and August separately. All the profiles are extremely close (by up to 10 ppbv) from the surface up to 400hPa, contrary to the upper troposphere, where the differences between the RR and the NR reach up to 150 ppbv at 200 hPa. We can see all along

the studied period that the NR exhibits greater ozone concentrations than the RR in the upper troposphere (between 200 and 500 hPa). The three ARs are closer to the NR than the RR, especially for S4+S5P_AR. However, this behaviour is completely different in the mid and lower troposphere: the NR presents on average lower ozone values than the ARs and the RR below 780, 550, 750, and 900 hPa for JJA, June, July and August, respectively, and the ARs overestimate both the RR and the NR.



This overestimation can also be seen by studying the MBE profile over the period (Fig. 7a). The bias for the three ARs is about 20% smaller than that of the RR in the upper troposphere, similar but with opposite sign in the mid troposphere and about 10% greater than to that of the RR in the lower troposphere. The RMSE (Fig. 7b) is up to 20% lower for the ARs than for the RR in the mid-to-upper troposphere (200 to 600 hPa) but slightly greater below this level (up to 5%). The mean

skill score profile over the period (Fig. 7c) shows a reduction of the RMSE in the mid-to-upper troposphere (above 600 hPa) reaching more than 30% above 450 hPa. Regarding the correlation coefficient difference between the ARs and the RR (Fig. 7d), S5P_AR presents a positive value in all the troposphere, especially noticeable in the upper stratosphere; however, S4_AR and S4+S5P_AR present positive values only from the upper part of the lower troposphere to the upper troposphere (200 to 700 hPa), but they are greater than those of S5P_AR between 300 and 600 hPa.

According to the tropospheric profiles analysis presented above, we selected three levels that will be more extensively validated in the upper (200 hPa), mid (500 hPa) and upper part of the lower troposphere (700 hPa). Other intermediate levels have also been studied (not shown) but these three levels are the most helpful to explain the results obtained in this OSSE.

As a first approach to analyse these three levels, we present the ozone averaged over the European domain time series for the RR, the NR and the three ARs in Fig. 8. At 200 hPa, the ARs (ranging from 200 to 150 ppbv along the period) are halfway

between the RR (ranging from 180 to 100 ppbv) and the NR (about 250 ppbv), while at 500 hPa the ARs are similar to the NR (around 70 ppbv), indicating an impact of the synthetic satellite data in the data assimilation process for these two levels. However, for the level 700 hPa, the ARs (around 60 ppbv) slightly overestimate the NR (around 55 ppbv), likely due to a small downward contribution of the upper levels. This overestimation is discussed in Sect. 7.

The next sections show more detailed results from these three levels. Figure 9 presents the three ARs, the NR and the RR

ozone fields averaged for the summer 2003 over the European domain at 200 hPa, 500 hPa and 700 hPa. To analyse these results in detail, we calculate the MAE (Fig. 10), the skill score (Fig. 11) and the correlation coefficient (Fig. 12) for the three levels.

## 6.1    At 200 hPa

Figure 9 (left column) shows changes clearly visible in the ozone AR fields at 200 hPa with a significant increase of ozone in

the northern part of the European domain, which means the AR is closer to the NR than the RR. At this level, the assimilated ozone fields for S4_AR, S5P_AR and S4+S5P_AR present similar patterns comparable with those of the NR, showing an equivalent added value.

Figure 10 (left column) shows the MAE fields averaged over the studied period at 200 hPa for S4_AR, S5P_AR, S4+S5P_AR and the RR. The MAE fields for the three ARs are similar but much smaller than that of the RR. In general, the MAE is smaller

in the northern part than in the southern part of the European domain. This is especially marked for the S4_AR and S4+S5P_AR compared to S5P_AR showing a slightly higher added value of S4 data at this level. The spatially averaged MAE time series of the three ARs (Fig. 10 - left column bottom) are lower than that of the RR all along the studied period. The MAE of the RR goes from 35% up to 65%. Conversely, the MAE of the three ARs is much smaller ranging between 20% and 45%, with an average of about 30%. The simultaneous assimilation of both S4 and S5P data provides a slightly smaller MAE than S4_AR,



which in turn is smaller than S5P_AR. This demonstrates the benefit of the assimilation of the satellite data at 200 hPa, in particular the synergy of both S4 and S5P data.

Figure 11 (left column) shows the mean skill score fields and time series for the three ARs over the studied period at 200 hPa. There is a net improvement in the full domain in terms of skill score for the three ARs, with S4+S5P_AR presenting slightly greater skill score values than S4_AR, and in turn S4_AR performs better than S5P_AR. This is clearly seen in Fig. 11 (left column bottom) where the skill score is increasing over time for the three ARs reaching more than 0.5 for S4+S5P_AR at the end of the period.

Figure 12 (left column) shows the correlation coefficient mean fields and the correlation coefficient histogram for the three ARs and the RR with NR at 200 hPa over the studied period. The behaviour of the NR, RR and AR is similar in terms of spatial distribution which is clearly seen in the correlation coefficient fields at this level. However, there is an improvement in the correlation histograms for the ARs (Fig. 12 - left column bottom). In addition, at 200 hPa, the histograms of the ARs are narrower than that of the RR, and the peak of the correlation histogram goes from less than 0.8 (RR) to 0.9 (ARs).

## 6.2   At 500 hPa

At 500 hPa, the assimilated ozone fields for S4_AR, S5P_AR and S4+S5P_AR present similar patterns and are closer to the NR than the RR (Fig. 9 - middle column). However, there is an overestimation of ozone in the South-East part coming from the assimilation of S4 data as one can see for S4_AR and S4+S5P_AR. We discuss this overestimation in Sect. 7.

As shown in Fig. 10 (middle column), the MAE fields for the three ARs present small values over all the European domain (around 10%) which are similar but much smaller than those of the RR. Greater MAE values are located in the North-West part of the European domain reaching up to 20% in particular for S5P_AR, and in the South-West part of the European domain with values up to 22% for the three ARs, but still smaller than the MAE RR values. In addition, the S4_AR and S4+S5P_AR exhibit MAE values reaching up to 22% in the South-East part of the European domain consistent with the overestimation found in the ozone fields. Regarding the temporal evolution (Fig. 10 - middle column bottom), the MAE for the three ARs is stable during the whole studied period, ranging from 10% to 15%, while the MAE for the RR increases during July and August, reaching more than 25%. The MAE for S4+S5P_AR is similar to that of S4_AR, but slightly smaller than that of S5P_AR. These results show the benefit of the assimilation of the either S4 or S5P satellite data at 500 hPa, but the synergy between the two instrument observations does not improve the analysis.

In the northern part of the European domain, the skill score for the three ARs shows greater values than in the southern part of the European domain (Fig. 11 - middle column). For the particular case of S4+S5P_AR and S4_AR, a negative skill score is found in the South-East, associated to the ozone overestimation mentioned above. In June, the skill score shows a significant variability while, in July and August, the mean skill score value is positive and increases reaching a stable value of 0.4 from mid-July to the end of August as shown in Fig. 11 (middle column bottom).

The three ARs improve the correlation coefficient field at 500 hPa (Fig. 12 - middle column) when compared to the RR, with a small advantage for S4+S5P_AR and S4_AR with respect to S5P_AR values. From the histogram (Fig. 12 - middle column





bottom), a clear improvement in the correlation coefficient provided by the assimilation of both S4 and S5P data is shown. The histogram maximum goes from 0.3 (RR) to 0.55 (for S5P_AR) and 0.65 (for S4_AR and S4+S5P_AR).

### 6.3 At 700 hPa

At 700 hPa, the assimilated ozone fields S4_AR, S5P_AR and S4+S5P_AR (Fig. 9 - right column) present similar patterns and

show values greater than those of the RR and the NR, consistent with the time-series in Fig. 8 (bottom).

The MAE presented in Fig. 10 (right column) reflects this fact mostly in the southern and eastern parts of the European domain where the MAE values of the ARs (S4_AR, S5P_AR and S4+S5P_AR) are greater than those of the RR. However, this is less pronounced for S4_AR and S4+S5P_AR. Conversely, in the remaining European domain, the MAE of the ARs is smaller (∼10%) than the MAE of the RR (∼15%) showing an improvement of the different ARs. The MAE time series (Fig.

10 - right column bottom) is consistent with the mean fields except for August when the MAE of the ARs becomes smaller than the MAE of the RR with still some variability.

The skill score mean fields (Fig. 11 - right column) for the three ARs is clearly separated into two parts: one with positive values (North-West part of the European domain coloured in red) and the other part with negative values (southern and eastern parts of the European domain coloured in blue). The positive skill score region indicates a RMSE reduction of the ARs with

respect to the RMSE of the RR reaching up to 30%. The negative skill score pattern found in the southern and eastern parts of the domain is consistent with the ozone overestimation and the greater MAE. Note that for August (Fig. 11 - right column bottom), the ARs skill score becomes positive following the behaviour of the MAE time series.

Like at the higher levels (200 and 500 hPa), the three ARs improve the correlation coefficient field at 700 hPa (Fig. 12 - right column) compared to the RR values. This shows an improvement in terms of patterns almost all over the full European

domain. The correlation coefficient values for the ARs range between 0.4 and 0.8 whereas those of the RR range from 0.2 to 0.75. The improvement is also highlighted by the peaks location of the histogram of the full set of data (Fig. 12 - right column bottom), which has increased from 0.55 (RR) to 0.6-0.7 (ARs).

### 7 Discussion

In Sect. 6, we presented the distribution of the mean ozone fields, the mean absolute error, the skill score (equivalent to RMSE

reduction) and the correlation coefficient for 200, 500 and 700 hPa. The metrics were chosen to evaluate the added value of the assimilation of the S4 and S5P data in terms of absolute error, improvement of the error, and agreement with the NR. The added value of the S4 and S5P data is well characterized when all the three metrics consistently show an improvement. The correlation coefficient of the ARs is always greater than the RR one for the three levels. However, the improvement in terms of MAE and skill score depends on the assimilation run time and/or the region studied for each level.

At 200 hPa, the results obtained from these metrics for the three ARs are consistent during all the studied period (JJA) and the whole domain. Compared to the RR, we find a reduction around 30% and up to 50% for the MAE and for the RMSE, respectively. Moreover, there is an increase of the correlation coefficient of 0.1 with a narrower histogram. Clearly,



the assimilation of satellite data brings ozone information at this level, which is in line with the vertical sensitivity of the satellite data used in this work.

Conversely, for the levels 500 and 700 hPa, the results show on average an added value but not for all the studied period and/or the whole domain. Regarding the studied period, an added value is shown for July and August at 500 hPa, and for

August at 700 hPa. This delay is likely due to the information from the levels above that impact these lower levels. For July and August, we find a reduction of the MAE for the ARs of more than 10% with respect to the RR and a skill score value reaching more than 0.4 at 500 hPa. At this level, we obtain an increase of the correlation coefficient from 0.25 to 0.35 when compared to the RR for the whole period. At 700 hPa and for August, the reduction of the MAE for the ARs compared to the RR is around 5% and the skill score is around 0.2. At this level and for the whole period, the correlation coefficient of the ARs

increases between 0.05 and 0.15 compared to the RR one.

A detailed analysis of these results at 500 hPa shows that the improvement of the skill score during July and August is due to the fact that the RMSE of the RR increases during this period (as seen in the MAE) while the RMSE of the ARs is stabilized by the impact of the assimilation of both S4 and S5P data (not shown). A similar behaviour is seen at 700 hPa but for August. This indicates that there is not a clear direct impact of S4 and S5P data at these two levels likely due to the low sensitivity of

these two instruments in the lower troposphere.

An ozone overestimation occurs in the South-East corner of the European domain. This is well seen at 500 hPa for the assimilation runs containing S4 data and more pronounced at 700 hPa for the three ARs. To better understand this fact, we calculate the zonal mean ozone profiles during the summer 2003 for four different latitudinal bands and the South-East corner, which are presented in Fig. 13. One can clearly see that the RR ozone profile shape is similar to the NR one for high latitudes,

but significantly different for lower latitudes, especially for the South-East corner. The gradient of the ARs profiles appears to be in line with that of the RR.

In our assimilation system, we used a B-matrix which does not evolve in time with the variance proportional to the model profile as described in Sect. 2. The assimilation of eigenvectors can be understood as the assimilation of several partial columns (6 in our case) with associated vertical sensitivity represented by the transformed AKs. From Fig. 4, one can conclude that

the shape of the AKs in the troposphere is very similar, meaning that there is at most a tropospheric column information, with higher sensitivity in the upper troposphere. If we consider a single model grid point, the assimilation process spreads the eigenvectors information on the vertical (from the stratosphere -with very high values- to the troposphere -with very low values-) by calculating an increment profile that minimizes the distance to the assimilated data. The assimilated profile is then a shifted profile resulting from the sum of the background profile and the calculated increment profile, and is highly dependent on the

background profile shape and the B-matrix. In particular, when the background profile shape is not following the NR shape, this could sometimes lead to a significant over/under estimation in the levels where the assimilated data has a low sensitivity. The S4 and S5P satellite sensors have a higher sensitivity in the upper troposphere and a lower sensitivity in the lower troposphere for ozone. In our case, the NR ozone concentration is greater than the background in the upper levels resulting in an overestimation of ozone in the lower levels due to the fact that the distribution of the ozone information is governed by the background profile

shape and the AKs, which have much higher sensitivity in the upper troposphere compared to the lower troposphere. Notice





that the vertical extent of the impact of GBS data assimilation is limited by the background error vertical correlation which lengthscale has been set to one model level, and therefore the use of this data does not compensate this effect at 700 hPa and the height levels above.

## 8 Conclusions

We performed assimilation runs with synthetic data that mimic S4 and S5P satellite observations over Europe and during the period of the summer 2003. The reference run was performed with the assimilation of the simulated GBS ozone data, using the same approach that is commonly used in an operational AQ forecast system. We analysed the troposphere, with a focus on the levels at 200, 500 and 700 hPa.

For the development of the ozone OSSE, an efficient interface to ozone observations has been used. More specifically, one
of the innovations of this work is the generation of the ozone profile information in the form of leading eigenvectors of the radiative transfer code. This represents a very efficient and convenient interface between the retrievals and the data assimilation system. The use of this approach has been validated in this work. In addition, we have shown the importance of correctly adding the representativeness uncertainties into the observation error covariance matrix.

The OSSE that we have set up is as little overoptimistic as possible to ensure the robustness of the results. The retrieved
ozone profiles of S4 and S5P were obtained using the same spectral range (300-320 nm), and stored in the form of synthetic observations (six leading eigenvectors). Note that the instrumental characteristics chosen for the retrievals do not use all the capacities of S5P and S4 but are assumed to be consistent with the actual characteristics of S5P (which is already flying) and the future S4 missions. The nature run is composed of two different CTMs, LOTOS-EUROS and TM5, and is built by merging the ozone profiles from the former for the boundary layer with the ones from the latter from the free troposphere to
the stratosphere. A different model (MOCAGE) was used to perform the reference and assimilation runs, in order to avoid the identical twin problem. The diagonal of the background error covariance matrix is proportional to the model ozone profiles and does not evolve in time.

Under these conditions, we show that both S4 and S5P bring information from the upper troposphere to the middle troposphere. The maximum added value is above 500 hPa. As expected, the assimilation of both S4 and S5P ozone shows better
results than the reference run and is closer to the nature run up to these altitudes (in terms of mean absolute error, skill score and correlation coefficient). At 200 hPa there is a reduction of MAE from more than 60% to a more stable MAE of about 30% (for S4+S5P_AR). There is also a reduction in the RMSE (skill score) of the ARs of up to 50% compared to the RR and a better correlation with the NR.

The behaviour of the assimilation runs S4+S5P_AR and S4_AR is quite similar in terms of MAE, reduction of RMSE (skill
score) and correlation coefficient, and slightly better than the S5P ozone assimilation (S5P_AR) between 200 and 700 hPa. However, there is no significant difference between the added value given by the GEO S4 and the LEO S5P. This is likely due to the fact that there is no diurnal cycle of ozone above the boundary layer, so the information provided by a LEO is still adequate to constrain the model.





The outcome of our study is a result of the OSSE design and the choice into the components of the entire system: the synthetic observation characteristics and uncertainty estimates, the assimilation approach, the treatment of the observations in the assimilation, and the modelling characteristics. Under these conditions, we show that a significant benefit from the S4 and S5P observations is found in the middle troposphere (200 - 500 hPa). Moreover, at 200 hPa, the S4 and S5P increment

values obtained are larger than at the lower troposphere, showing the added value obtained at this level from S4 and S5P ozone. However, we did not find any significant impact at the lower troposphere (neither at the surface - not shown in the present study) from any of the experiments based only on these UV ozone profile observations. From these observations, we obtain about one piece of information in the troposphere, with a larger sensitivity in the free troposphere compared to the boundary layer. These results confirm that the use of observations derived from the UV is of limited use to obtain the ozone distribution

within the boundary layer, required for air quality. The assimilation of retrievals of total column ozone from S5P real data is currently being tested and appears to have a small impact in the CAMS analysis (Inness et al., in review, 2018). A way to overcome this issue is to combine observations from various wavelength ranges, such as UV and Infrared or UV and Visible. An example will be the combination of observations from S4 UVN and Infrared Sounder (IRS) both on board MTG, whose study is out of the scope of this paper.

*Acknowledgements.* Support for this work came partly from the ESA funded project "Impact of Spaceborne Observations on Tropospheric Composition Analysis and Forecast" (ISOTROP–ESA contract number 4000105743/11/NL/AF). W. Lahoz acknowledges support from an internal project from NILU. S. Quesada-Ruiz, J.-L. Attié, P. Ricaud, L. El Amraoui, and W. Lahoz acknowledge support from the RTRA/STAE. J. Kujanpää and J. Tamminen acknowledge support from the Academy of Finland (project no. 267442).




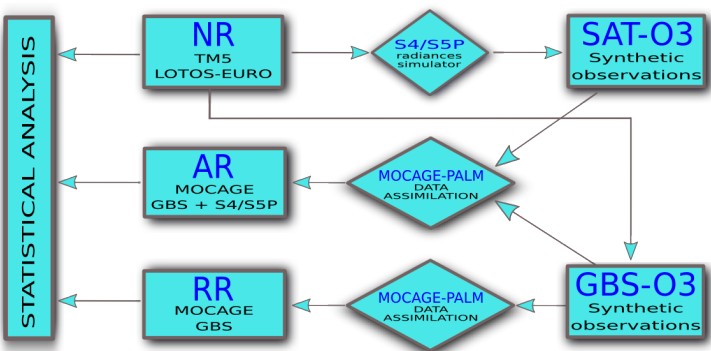

**Figure 1.** Diagram of the observing system simulation experiment (OSSE) components for ozone, including ground-based stations (GBS) measurements, and synthetic satellite observations from Sentinel-4 (S4) and Sentinel-5P (S5P) instruments.



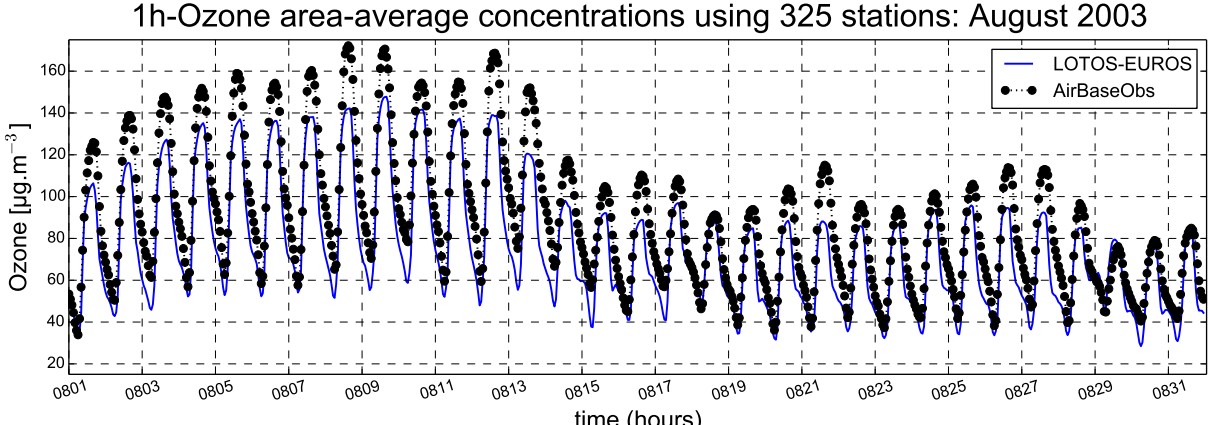

**Figure 2.** Surface ozone data (in µg.m$^{-3}$) averaged over 325 stations during August 2003 as described by LOTOS-EUROS model (blue line) and compared to Airbase database ground-based stations measurements (black dotted line and circles).





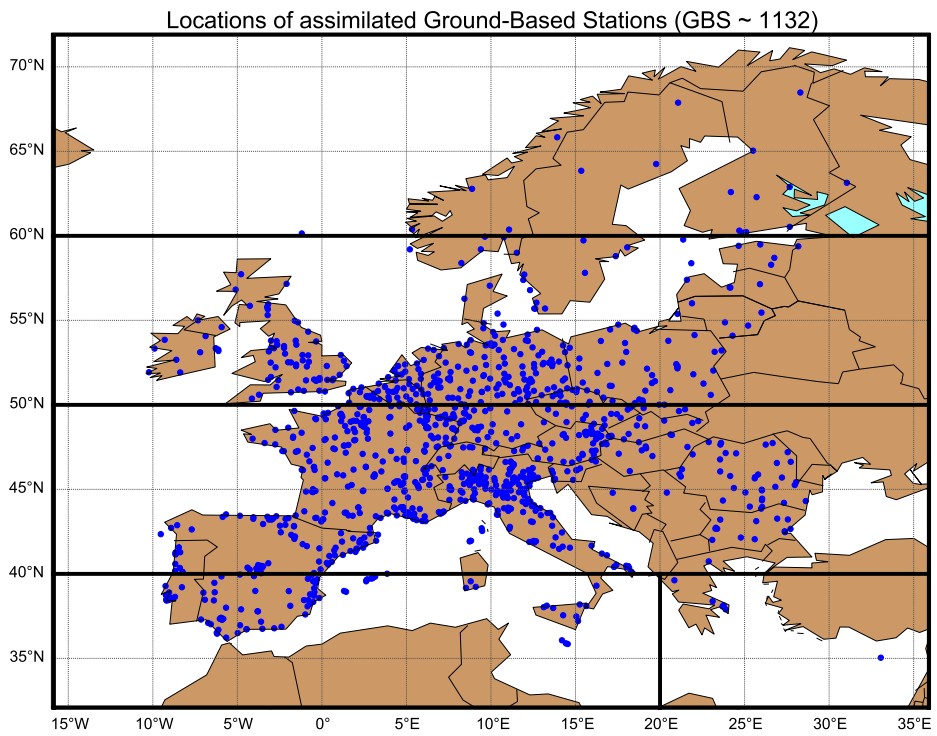

**Figure 3.** Ground-based stations (GBS) location (blue dots). We assimilated the simulated ozone concentration corresponding to the location of these 1132 ground-based stations from the Airbase database.





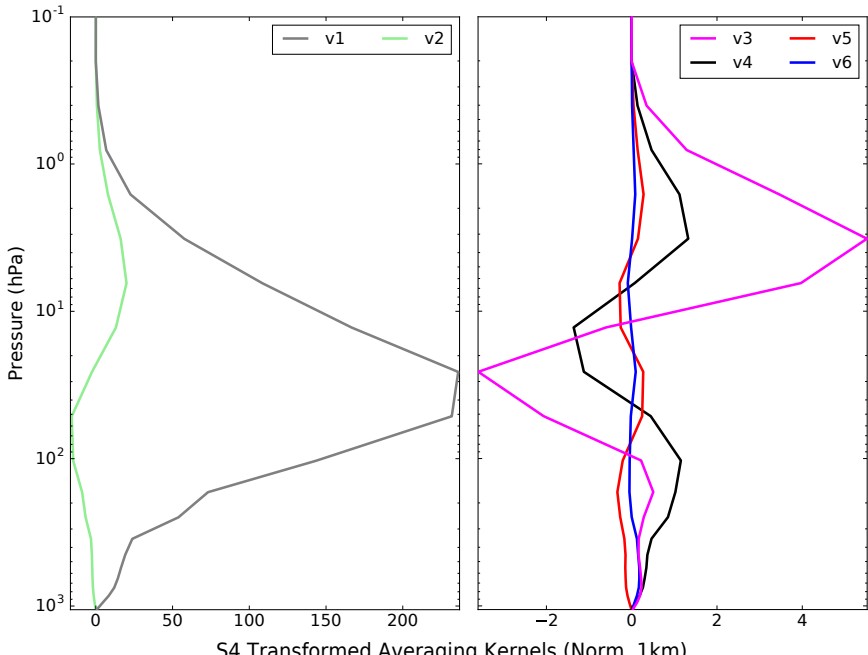

**Figure 4.** S4 ozone retrieval mean transformed averaging kernel for summer 2003 over the European domain for the six first leading eigenvectors. Left panel: v1 (grey line) and v2 (green line). Right panel: v3 (magenta line), v4 (black line), v5 (red line) and v6 (blue line). Note that x axis scale change between the two panels, and the AKs are normalized to 1 km by dividing them by the dz (difference between the top and bottom heights of the layer) corresponding to each layer.





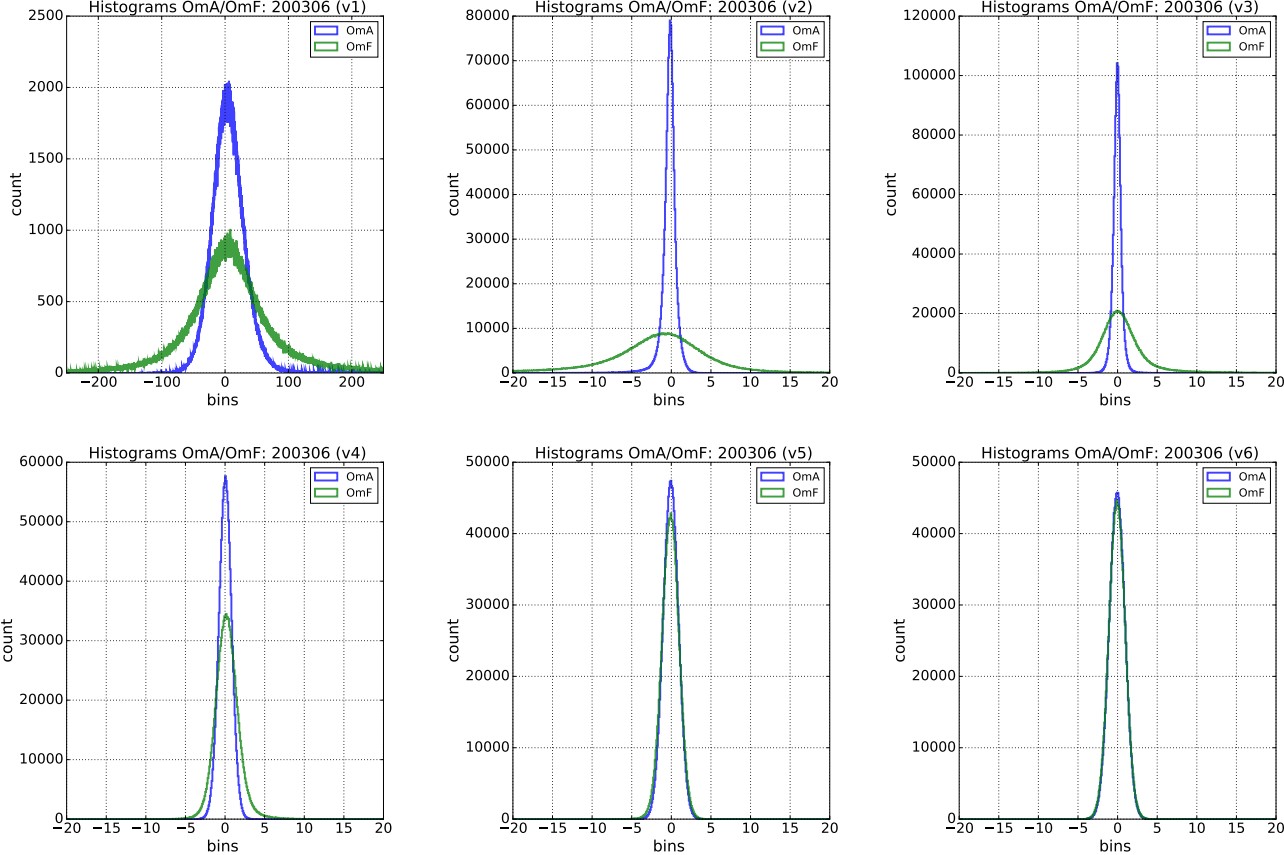

**Figure 5.** Histograms of Observation minus Analysis (OmA) -blue lines- and of Observation minus Forecast (OmF) -green lines- for the first leading eigenvectors v1 to v3 (from left to right, top) and v4 to v6 (from left to right, bottom) for the assimilation of S4 during the month of June 2003. Note that the y axis is variable for each eigenvector, while the x axis is different for v1.



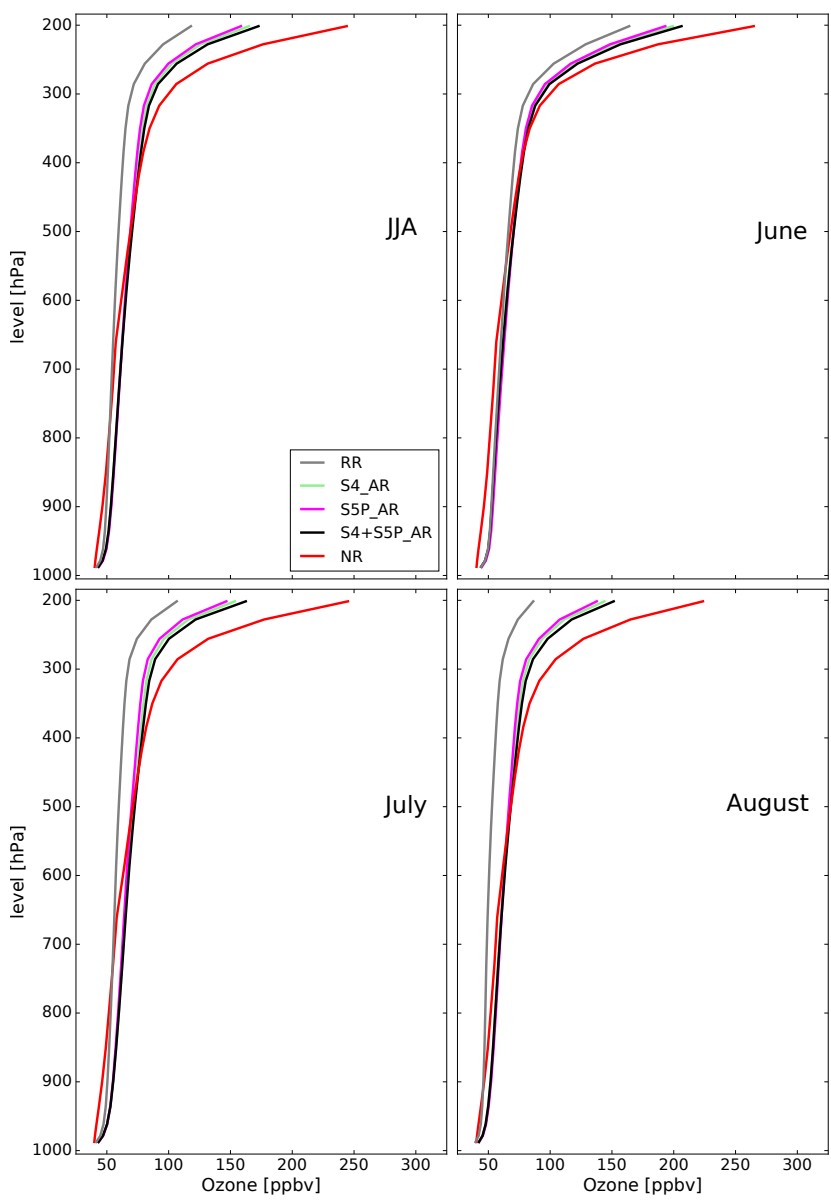

**Figure 6.** Ozone profiles (in ppbv) averaged for the summer 2003 (JJA) and June (top row), July and August (bottom row) over the European domain, as simulated from the RR (gray lines), the S4_AR (green lines), the S5P_AR (magenta lines), the S4+S5P_AR (black lines) and the NR (red lines).



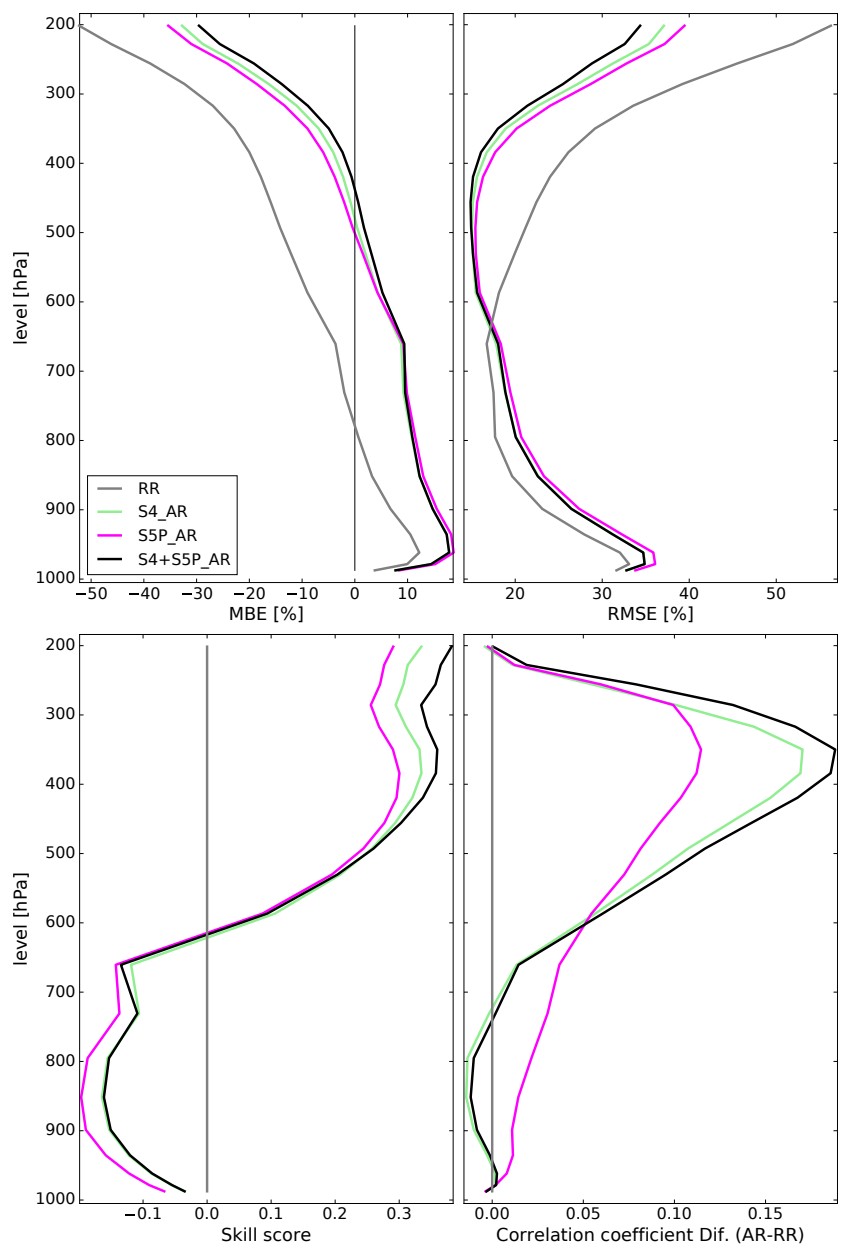

**Figure 7.** Ozone MBE (in %), RMSE (in %), Skill score, and Correlation coefficient difference (ARs-RR) mean profiles for summer 2003 (JJA) over the European domain (from left to right and from top to bottom, respectively) corresponding to the RR (grey lines), the S4_AR (green lines), the S5P_AR (magenta lines), and the S4+S5P_AR (black lines).





**Figure 8.** Ozone time-series (in ppbv) averaged for summer 2003 over the European domain at 200, 500 and 700 hPa (from top to bottom) as provided by the RR (grey lines), the S4_AR (green lines), the S5P_AR (magenta lines), the S4+S5P_AR (black lines) and the NR (red lines). The horizontal axis represents the studied period, from 01/06/2003 to 31/08/2003, in steps of 3 hours.





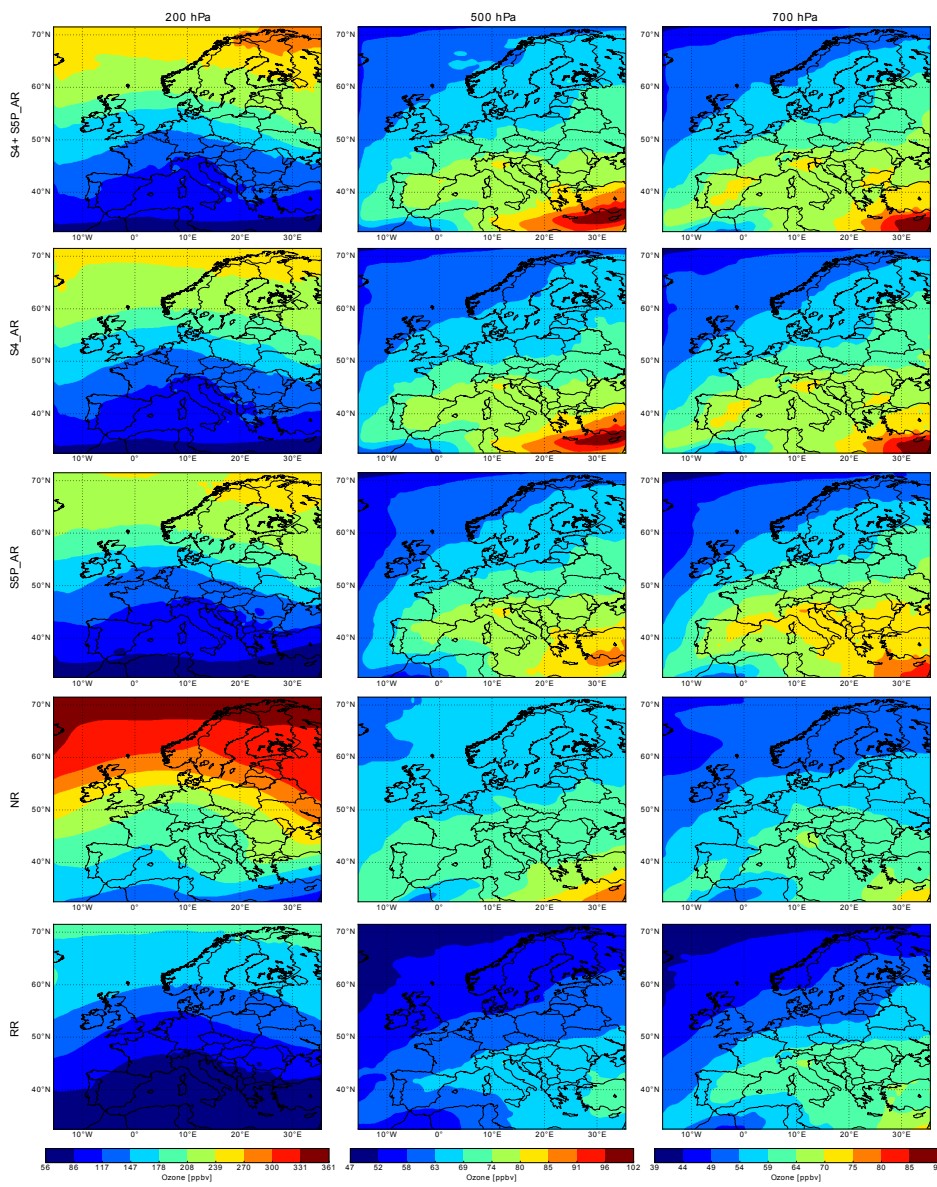

**Figure 9.** Mean ozone fields (in ppbv) for the summer 2003 over the European domain at 200 hPa (left column), 500 hPa (middle column) and 700 hPa (right column), as simulated by the S4+S5P_AR, the S4_AR, the S5P_AR, the NR, and the RR (from top to bottom rows). Note that the colour bars for the three levels are different. Red (respectively blue) end of the colour scale represents relatively large (respectively small) ozone values.





**Figure 10.** Mean absolute error (MAE) fields (in %) for the summer 2003 over the European domain at 200 hPa (left column), 500 hPa (middle column) and 700 hPa (right column), and for the S4+S5P_AR, the S4_AR, the S5P_AR, and the RR (from top to bottom, first to fourth rows) compared to the NR. Note that the colour bars for the three levels are different. Red (respectively blue) end of the colour scale represents relatively large (respectively small) MAE values. Bottom row represents the MAE (in %) time series for summer 2003 over the European domain at 200 hPa (left column), 500 hPa (middle column) and 700 hPa (right column) as provided by the RR (grey lines), the S4_AR (green lines), the S5P_AR (magenta lines), and the S4+S5P_AR (black lines).





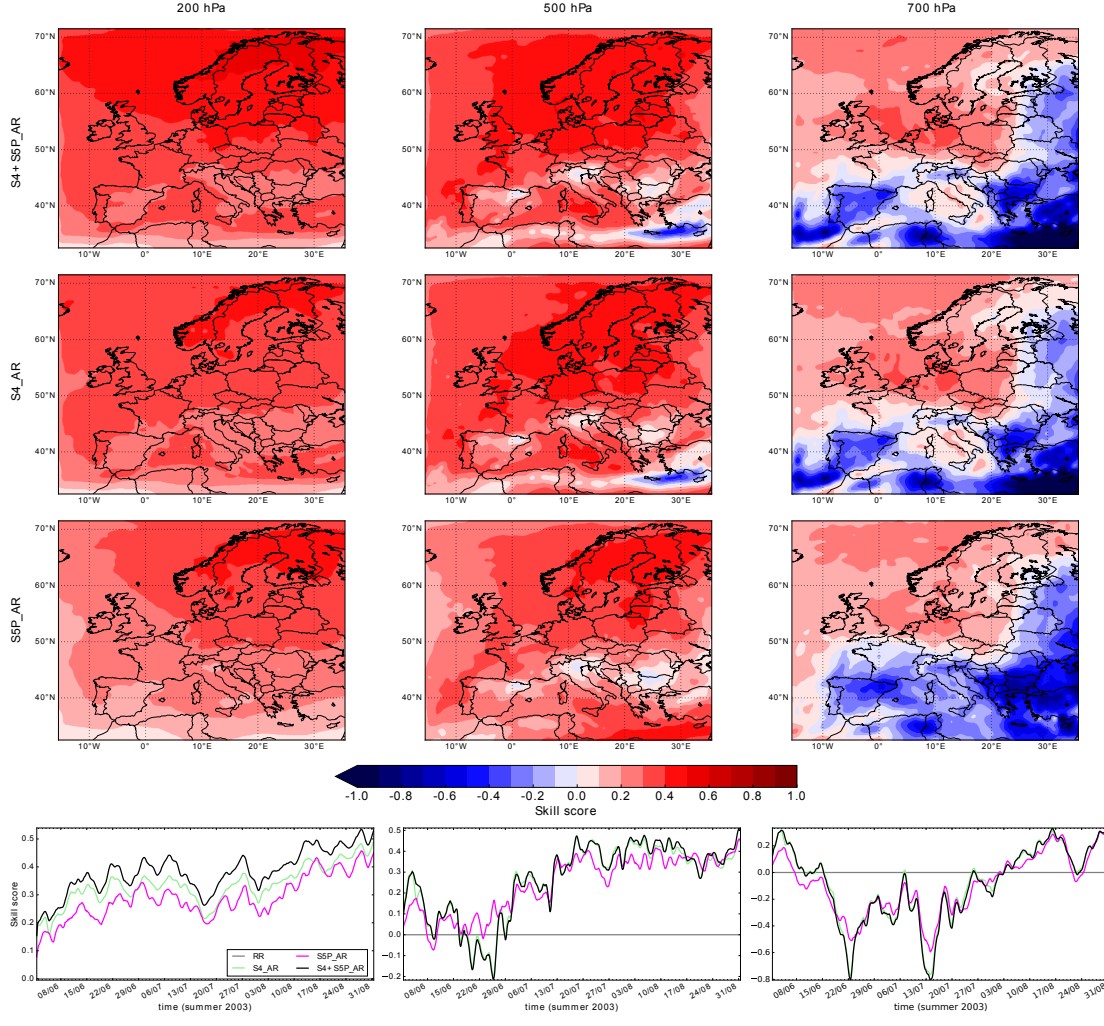

**Figure 11.** As Figure 10 but for the Skill score maps and time series. Note that there is no Skill score map for RR because the skill score is calculated with respect to the RR, so it is equal to 0. Red (respectively blue) colour scale is associated with positive (respectively negative) skill score values.

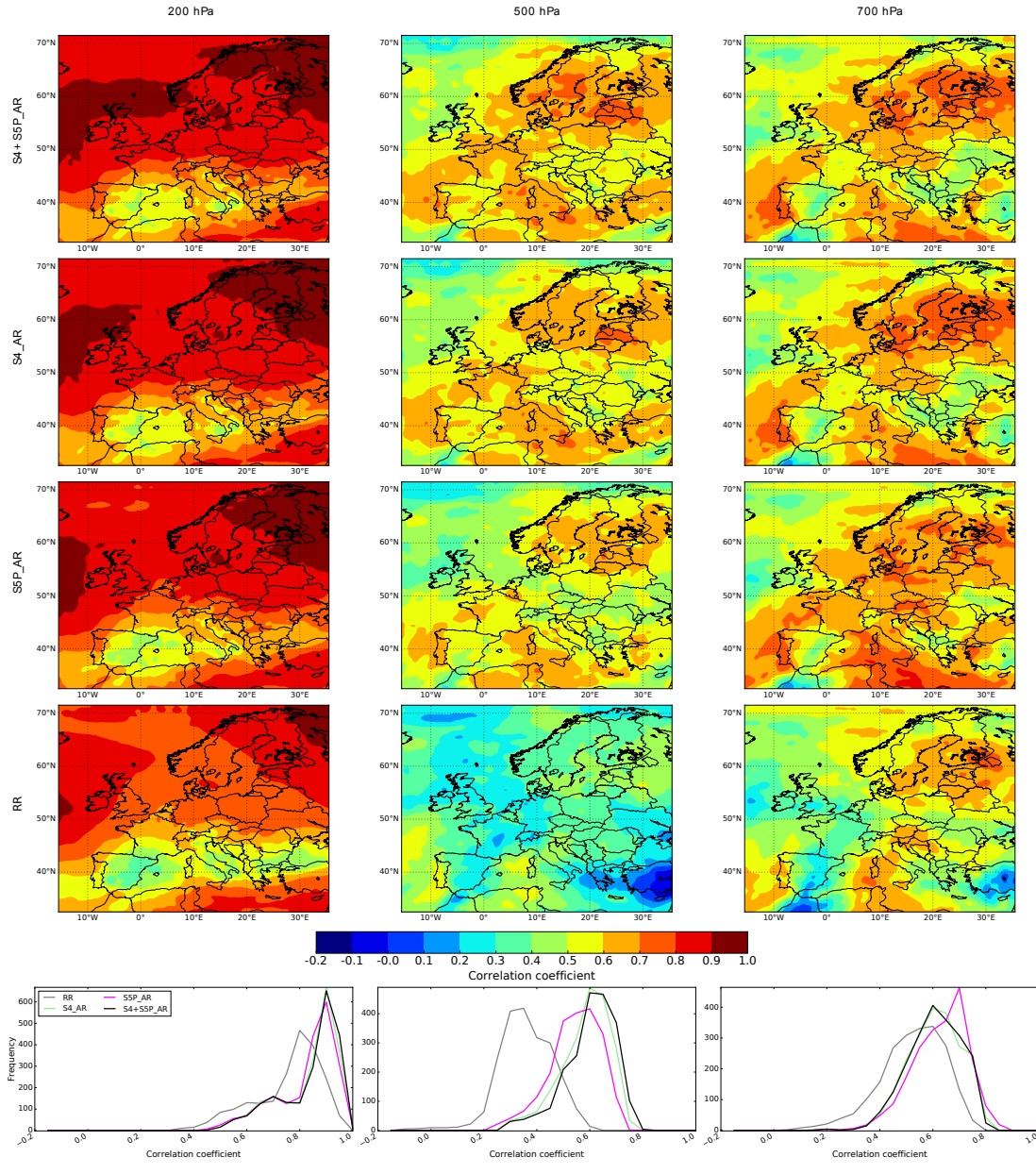

**Figure 12.** As Figure 10 but for the Correlation coefficient maps and histograms. Note that the colour bars are the same for the three levels.







**Figure 13.** Ozone profiles (in ppbv) averaged for the summer 2003 (JJA) by geographical zone: over the European domain and zones Z1 [60-72ºN] and Z2 [50-60ºN] (top row, from left to right), and zones Z3 [40-50ºN], Z4 [32-40ºN], and Z4B [32-40ºN, 20-36ºE] (bottom row, from left to right), as simulated from the RR (grey lines), the S4_AR (green lines), the S5P_AR (magenta lines), the S4+S5P_AR (black lines) and the NR (red lines). These zones are identified with black boxes over Fig. 3.





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
