# Peer review of "Benefit of ozone observations from Sentinel-5P and future Sentinel-4 missions on tropospheric composition"

_Atmospheric Measurement Techniques, 2018_

## Referee Comment (RC1) · Anonymous Referee #1 · 8 Mar 2019

General comment: In this paper, Samuel Quesada-Ruiz et al. evaluated the benefits of assimilating ozone observations from sentinel-5P and sentinel-4 in the MOCAGE-PALM system using a synthetic study in the European domain. They quantified the improvements and deterioration of O3 profile results at levels of 200, 500, and 700hPa by adding the new satellite observations. Since the real ozone profiles can not be well known for the real measurements, the synthetic study with CTM simulations of real atmosphere is a state-of-art way for the research purpose. In general, the scientific topic is meaningful, research method is novel, and presentation is quite concise.

I have one general concern. In the simulations of S4 and S5P ozone observations,

the transformed AK, which is derived from the DISAMAR inversion package, is used to convert the true ozone profile from the nature run to the measured ozone profile. The AK is not only as a function of atmospheric statement and measured geometries, but also depends on settings of optimal estimations, e.g. covariance of a-priori and measurement uncertainties. And the measured ozone profiles calculated with the AK can further impact the final ozone profile results from the assimilation run. The benefit of S4 and S5P ozone observations is concluded based on the assimilation run. Therefore the conclusion might be specifically for the DISAMAR inversion package. The benefit might be different if other inversion algorithms are applied to the ozone retrievals of S4 and S5P observations. Considering this, the authors should clarify that the conclusion is based on the DISAMAR inversion package and might be different for other algorithms of ozone retrievals in the abstract and conclusion part.

Specific comments:

1) The abbreviations of nature run, assimilation run, and reference run might not be needed. It is easier for readers to understand the paper if the original words are written in the manuscript.

2) P3, L3-4: The data assimilated to the MOCAGE-PALM system should not be the ozone data simulated from the nature run. As I understand, the data should be simulated ozone observations with the ozone data simulated from the nature run. Please check the sentence.

3) P4, L29: The free run is not explained in the paper. Please check.

4) P5, L32: "The simulated ozone observations from GBS" should be based on the nature run results and assimilated into the MOCAGE-PALM. Please check the sentence.

5) Section 4.2.1: Since the spectral analysis of ozone is not applied in the synthetic study, how do you consider the uncertainty of spectral analysis for satellite observations?

---

## Referee Comment (RC2) · Anonymous Referee #2 · 28 May 2019

In this study, Quesada-Ruiz et al. conducted an Observing Simulated System Experiment (OSSE) in order to assess the benefit of future ozone data from individual or combined use of GEO (Sentinel-4) and LEO (Sentinel-5P) satellite observations on tropospheric ozone composition. This OSSE, which focused over Europe during the summer 2003 period, consisted in the following two main steps: (1) assimilating S4 and S5P synthetic ozone profile data simulated by the DISAMAR inversion package using LOTOS-EUROS and TM5 3D-CTM fields as input, and (2) comparing the assimilation results to a reference run based on the assimilation of simulated ozone data at a selection of 1132 AirBase stations. Results showed that S4 and S5P satellite data in the UV range clearly bring direct added value to the tropospheric ozone composition

in the middle troposphere (200-500 hPa). This study also confirmed the limited use of satellite observations in the UV for deriving the ozone distribution inside the boundary layer.

The manuscript is well written and clearly structured and the presented results are scientifically relevant. I recommend the paper for publication in AMT after addressing the following specific comments:

1/Page 4, lines 16-24: one important parameter which has a large impact on the analysis is the background error covariance matrix. The authors should further justify how they built this matrix and why it does not evolve with time. For instance, it would be also interesting to know whether the chosen variance and correlation lengths values come from prior sensitivity tests.

2/Page 5, line 17: It would help the reader to briefly summarize what were the evaluation results of TM5 ozone data against MOZAIC aircraft measurements in August 2003.

3/Page 5, lines 27-28: The bias between LOTOS-EUROS and the surface ozone measurements is about 10-20$\mu$g.m-3. Does this bias can be considered as a bad, fair or good agreement ? Given the fact that the selection of the nature run component is of great importance for the OSSE, what is the impact of this bias on the results of the study ?

4/Page 6, lines 26-29: Could you justify the choice of the SNR values for the solar irradiance and Earth radiance measurements ?

5/Page 12, lines 18-20: Pixels with cloud fraction greater than 0.05 have been discarded from the analysis. Does it mean that the methodology presented in the manuscript is only valid in clear-sky conditions ? Is a cloud treatment included in the DISAMAR package ? It would be useful to further comment on this cloud issue.

---

## Author Comment (AC1) · 10 Jul 2019

Interactive comment on

"Benefit of ozone observations from Sentinel-5P and future Sentinel-4 missions on tropospheric composition" by Samuel Quesada-Ruiz et al.

Anonymous Referee #1

We thank the anonymous reviewer for his/her valuable remarks and we answer point to point to the comments.

General comment:

In this paper, Samuel Quesada-Ruiz et al. evaluated the benefits of assimilating ozone observations from sentinel-5P and sentinel-4 in the MOCAGE- PALM system using a synthetic study in the European domain. They quantified the improvements and deterioration of O3 profile results at levels of 200, 500, and 700hPa by adding the new satellite observations. Since the real ozone profiles can not be well known for the real measurements, the synthetic study with CTM simulations of real atmosphere is a state-of-art way for the research purpose. In general, the scientific topic is meaningful, research method is novel, and presentation is quite concise.

I have one general concern. In the simulations of S4 and S5P ozone observations, the transformed AK, which is derived from the DISAMAR inversion package, is used to convert the true ozone profile from the nature run to the measured ozone profile. The AK is not only as a function of atmospheric statement and measured geometries, but also depends on settings of optimal estimations, e.g. covariance of a-priori and measurement uncertainties. And the measured ozone profiles calculated with the AK can further impact the final ozone profile results from the assimilation run. The benefit of S4 and S5P ozone observations is concluded based on the assimilation run. Therefore the conclusion might be specifically for the DISAMAR inversion package. The benefit might be different if other inversion algorithms are applied to the ozone retrievals of S4 and S5P observations. Considering this, the authors should clarify that the conclusion is based on the DISAMAR inversion package and might be different for other algorithms of ozone retrievals in the abstract and conclusion part.

Specific comments:

1) The abbreviations of nature run, assimilation run, and reference run might not be needed. It is easier for readers to understand the paper if the original words are written in the manuscript.

We will avoid all the acronyms that concern nature run, assimilation run and reference run throughout the manuscript, except for the figures.

2) P3, L3-4: The data assimilated to the MOCAGE-PALM system should not be the ozone data simulated from the nature run. As I understand, the data should be simulated ozone observations with the ozone data simulated from the nature run. Please check the sentence.

Right, we change the sentence as required by the reviewer.

3) P4, L29: The free run is not explained in the paper. Please check.

The free run has been removed from P4 L29. It is not required in the paper as the comparisons of the assimilation runs are based on the reference run.

4) P5, L32: "The simulated ozone observations from GBS" should be based on the nature run results and assimilated into the MOCAGE-PALM. Please check the sentence.

Yes, this is correct, the sentence is now changed.

5) Section 4.2.1: Since the spectral analysis of ozone is not applied in the synthetic study, how do you consider the uncertainty of spectral analysis for satellite observations?

The answer to the general concern and question 5 contains several elements: the radiative transfer code, the a-priori, the chosen uncertainty on the radiance and irradiance, absolute calibration of the instrument, and the representativity error. We will comment on each of these elements separately: - The DISAMAR radiative transfer / retrieval code was used. DISAMAR, which builds on the Doubling-Adding code of KNMI (DAK) has been compared extensively against other state-of-the-art radiative transfer codes, and generally the quality of the retrieval is not limited by which code is used, but much more so by the input parameters such as instrument noise. - The a-priori. Indeed, the retrieved profile will depend on the a-priori and a-priori covariance. However, as explained by Migliorini (2012), in contrast the assimilation results are not (or only weakly in the case of non-linearity) dependent on the a-priori in the retrieval because the averaging kernel effectively removes this a-priori dependence. So, also

the a-priori is not a limiting factor. - The assumptions for the SNR of the radiance and irradiance are described in the beginning of section 4.2. This is strongly wavelength dependent towards the UV because of the strong decrease of the signal. We believe that this choice of radiance noise is realistic for TROPOMI and we assume that the SNR for Sentinel-4 will be comparable. - The absolute calibration of the instrument is a much more serious issue and may lead to a systematic distortion of the profile shape. Unfortunately, such absolute calibration issues and instrument degradation can not be known before the instrument is in space. We assume that these errors are zero, or that this is a systematic feature which has been corrected for by soft calibration. In the case of TROPOMI (launced in October 2017) this turned out to be a major issue (after launch) and soft calibration is needed there. For other instruments, such as e.g. GOME-2, soft calibrations to correct for systematic biases and degradation have been applied with quite some success. - The representativity error: we believe that a nice aspect of our paper is the estimation of the representativity term, as described in section 4.2.1, table 1. In theory, the variance of the different eigenvector observations is 1.0, but in practice interpolation and coincidence errors increase the total observation error, and it is well understood that especially vector 1, which has a very small relative retrieval error, is mostly affected. So, we think we have an efficient representation of this term.

To conclude: we do not think the choice of the DISAMAR package or a-priori has a major impact on the results. Also, the SNR assumptions are realistic. In practice, the unknown calibration and degradation errors will be the most serious additional uncertainty on top of the uncertainties reported in our study.

S. Migliorini, On the Equivalence between Radiance and Retrieval Assimilation, Mon. Wea. Rev 140, 2012, DOI: 10.1175/MWR-D-10-05047.1

---

## Author Comment (AC2) · 10 Jul 2019

Interactive comment on

"Benefit of ozone observations from Sentinel-5P and future Sentinel-4 missions on tropospheric composition" by Samuel Quesada-Ruiz et al.

Anonymous Referee #2

In this study, Quesada-Ruiz et al. conducted an Observing Simulated System Experiment (OSSE) in order to assess the benefit of future ozone data from individual or combined use of GEO (Sentinel-4) and LEO (Sentinel-5P) satellite observations on

tropospheric ozone composition. This OSSE, which focused over Europe during the summer 2003 period, consisted in the following two main steps: (1) assimilating S4 and S5P synthetic ozone profile data simulated by the DISAMAR inversion package using LOTOS-EUROS and TM5 3D-CTM fields as input, and (2) comparing the assimilation results to a reference run based on the assimilation of simulated ozone data at a selection of 1132 AirBase stations. Results showed that S4 and S5P satellite data in the UV range clearly bring direct added value to the tropospheric ozone composition in the middle troposphere (200-500 hPa). This study also confirmed the limited use of satellite observations in the UV for deriving the ozone distribution inside the boundary layer. The manuscript is well written and clearly structured and the presented results are scientifically relevant. I recommend the paper for publication in AMT after addressing the following specific comments:

1/Page 4, lines 16-24: one important parameter which has a large impact on the analysis is the background error covariance matrix. The authors should further justify how they built this matrix and why it does not evolve with time. For instance, it would be also interesting to know whether the chosen variance and correlation lengths values come from prior sensitivity tests.

We agree on the importance of the B matrix in the assimilation system. Following the OSSE philosophy, we want to be as less overoptimistic as possible and this is why we decided to freeze our B matrix in time, that is, the same matrix is used for the whole assimilation period. The chosen correlation lengths and variance are commonly used in the MOCAGE-PALM assimilation system.

2/Page 5, line 17: It would help the reader to briefly summarize what were the evaluation results of TM5 ozone data against MOZAIC aircraft measurements in August 2003.

As recommended by the reviewer, a brief paragraph is added to summarize the comparison between TM5 and MOZAIC aircraft measurements.

3/Page 5, lines 27-28: The bias between LOTOS-EUROS and the surface ozone measurements is about 10-20$\mu$g.m-3. Does this bias can be considered as a bad, fair or good agreement ? Given the fact that the selection of the nature run component is of great importance for the OSSE, what is the impact of this bias on the results of the study ?

In the OSSE process we cannot use independent real data to evaluate the impact of the assimilation of the retrievals. We use a nature run, which is in fact just one possibility of the actual state of the atmosphere. The validation of the nature run against independent data is done to guarantee that this representation of the actual state of the atmosphere can be considered realistic.

Figure 2 shows that LOTOS-EUROS surface ozone is consistent with the Airbase database ground-based stations measurements, both in the diurnal cycle and in the temporal evolution of the min/max pics. Regarding the bias, and to stay objective as we can, we prefer to avoid to say bad, fair or good but we can say the bias represents about 10 to 18%. This is an information that could be extrapolated in the assimilated results, i.e., the assimilated results could have this bias compared to the reality and explain the low (or high) values from the assimilation.

4/Page 6, lines 26-29: Could you justify the choice of the SNR values for the solar irradiance and Earth radiance measurements ?

The signal-to-noise ratio is very strongly dependent on the wavelength in the region 300 to 320 nm. This dependence is mainly related to the signal, which very rapidly decreases towards the UV. The choice for the SNRs (50, 300, 1000) at 300, 310 and 320 nm is based on experience with existing sensors such as the ozone monitoring instrument (OMI), and on estimates and requirements for TROPOMI before launch (e.g. Veefkind et al., 2012). This is very close to actual post-launch estimates for TROPOMI, which are about (40, 316, 1000) for these wavelengths (N. Rozemeijer, KNMI, private communications). In between these values the SNR is interpolated.

Veefkind, J. P.; Aben, I.; McMullan, K.; Förster, H.; de Vries, J.; Otter, G.; Claas, J.; Eskes, H. J.; de Haan, J. F.; Kleipool, Q.; van Weele, M.; Hasekamp, O.; Hoogeveen, R.; Landgraf, J.; Snel, R.; Tol, P.; Ingmann, P.; Voors, R.; Kruizinga, B.; Vink, R.; Visser, H.; and Levelt, P. F. TROPOMI on the ESA Sentinel-5 Precursor: A GMES mission for global observations of the atmospheric composition for climate, air quality and ozone layer applications. rse, 120: 70–83. 2012.

5/Page 12, lines 18-20: Pixels with cloud fraction greater than 0.05 have been discarded from the analysis. Does it mean that the methodology presented in the manuscript is only valid in clear-sky conditions ? Is a cloud treatment included in the DISAMAR package ? It would be useful to further comment on this cloud issue.

Yes this study is only valid for clear sky conditions. We added this comment in the paper.

Clouds in DISAMAR are treated in an effective way in this study. The cloud is modelled as a Lambertian reflecting surface, specified by the cloud albedo (set to a fixed value of 0.8) and cloud pressure. Mixed scenes are modelled using the independent pixel approach, as weighted mean of a cloudy and cloud free part.

---

## Editor Decision (ED1)

Associated Editor's comments on the Authors' answers to the reviewers' comments on the AMT-Manuscript amt-2018-456, entitled:
**'Benefit of ozone observations from Sentinel-5P and future Sentinel-4 missions on tropospheric composition'**
**by Samuel Quesada-Ruiz et al.**

In general: The authors give satisfactory answers to the reviewers' comments and I strongly recommend submission of a revised version of the manuscript. I should, however, like to stress the following points, which are all about changes to the manuscript that are very advisable in my opinion, but are not specified in the authors' answers to the two reviewers' comments:

1) Editor's comments on the authors' answers to reviewer #1 (amt-2018-456-AC1)

Here changes to the manuscript in response to the general concern of reviewer #1 are required, i.e. the dependence of the usefulness on the choice of inversion package (DISAMAR) and assumption of parameters like assumed uncertainties?

2) Editor's comments on the authors' answers to reviewer #2 (amt-2018-456-AC2)

1/ If I understand this right the authors state here that their choice of a constant error covariance matrix is a conservative one. This needs to be explained.
Also: There should be a comment in the text of the revised manuscript stating that this choice was made and why.

3/ Do the authors mean that the bias constitutes 10 to 18% of the tropospheric ozone level? This needs to be mentioned in the revised version of the manuscript (page 5).

4/ What are the planned changes to the manuscript in response to this question of the reviewer?

---

## Author Response (AR2)

Associated Editor's comments on the Authors' answers to the reviewers' comments on the

AMT-Manuscript amt-2018-456, entitled:

**'Benefit of ozone observations from Sentinel-5P and future Sentinel-4 missions on tropospheric composition'**

**by Samuel Quesada-Ruiz et al.**

In general: The authors give satisfactory answers to the reviewers' comments and I strongly recommend submission of a revised version of the manuscript. I should, however, like to stress the following points, which are all about changes to the manuscript that are very advisable in my opinion, but are not specified in the authors' answers to the two reviewers' comments:

**We want to thank very much the Editor for his review and answer below point to point to his comments (in blue).**

1) Editor's comments on the authors' answers to reviewer #1 (amt-2018-456-AC1)

Here changes to the manuscript in response to the general concern of reviewer #1 are required, i.e. the dependence of the usefulness on the choice of inversion package (DISAMAR) and assumption of parameters like assumed uncertainties?

**In our first response to the reviewer #1, we answered to the general concern and to the question 5 together by adding a new section (section 4.2.2 Uncertainty of spectral analysis for satellite observations). However, to clarify this here, we only present the answer related to the general comments.**

**In section 4.2.2, the following paragraphs were added to respond to this general concern:**

**"i) The DISAMAR radiative transfer / retrieval code was used. DISAMAR, which builds on the Doubling-Adding code of KNMI (DAK) has been compared extensively against other state-of-the-art radiative transfer codes, and generally the quality of the retrieval is not limited by which code is used, but much more so by the input parameters such as instrument noise"**

**"To conclude, we do not think the choice of the DISAMAR package or a-priori has a major impact on the results. Also, the SNR assumptions are realistic. In practice, the unknown calibration and degradation errors will be the most serious additional uncertainty on top of the uncertainties reported in our study."**

In addition, in this section we discuss in detail the main input parameters in DISAMAR that more likely have an impact on the results (from line 11 to 28, pp 12):

"ii) The a-priori. The retrieved profile will indeed depend on the a-priori and a-priori covariance. However, as explained by Migliorini (2012) , in contrast the assimilation results are not (or only weakly in the case of non-linearity) dependent on the a-priori in the retrieval because the averaging kernel effectively removes this a-priori dependence. So, also the a-priori is not a limiting factor.

iii) The assumptions for the SNR of the radiance and irradiance are described in the beginning of Sect. 4.2. This is strongly wavelength dependent towards the UV because of the strong decrease of the signal. We believe that this choice of radiance noise is realistic for TROPOMI and we assume that the SNR for Sentinel-4 will be comparable.

iv) The absolute calibration of the instrument is a much more serious issue and may lead to a systematic distortion of the profile shape. Unfortunately, such absolute calibration issues and instrument degradation can not be known before the instrument is in space. We assume that these errors are zero, or that this is a systematic feature which has been corrected for by soft calibration. In the case of TROPOMI (launched in October 2017) this turned out to be a major issue (after launch) and soft calibration is needed there. For other instruments, such as e.g. GOME-2, soft calibrations to correct for systematic biases and degradation have been applied with quite some success.

v) The representativeness error. We believe that a nice aspect of our paper is the estimation of the representativeness term, as described in Sect. 4.2.1, Table 1. In theory, the variance of the different eigenvector observations is 1.0, but in practice interpolation and coincidence errors increase the total observation error, and it is well understood that especially vector 1, which has a very small relative retrieval error, is mostly affected. So, we think we have an efficient representation of this term."

Moreover, the use of DISAMAR has been highlighted in the abstract (Page 1 Line 4):

" … has been simulated using DISAMAR Inversion package ..."

 and a new sentence has been added at the end of the abstract Page 1 Lines 14,15:

"The outcome of our study is a result of the OSSE design and the choice into the components of the entire system"

Finally, the impact of the different OSSE components on the assimilation results was already mentioned in the conclusions (Page 20 Lines 15-17):

"The outcome of our study is a result of the OSSE design and the choice into the components of the entire system: the synthetic observation characteristics and uncertainty estimates, the assimilation approach, the treatment of the observations in the assimilation, and the modelling characteristics"

2) Editor's comments on the authors' answers to reviewer #2 (amt-2018-456-AC2)

1/ If I understand this right the authors state here that their choice of a constant error covariance matrix is a conservative one. This needs to be explained.

Also: There should be a comment in the text of the revised manuscript stating that this choice was made and why.

**Different techniques could be applied to optimize the error covariance matrix (B matrix). The use of these techniques would imply varying the B matrix along the assimilation experiment and would eventually result in an improved analysis. However, we considered a fixed B matrix which is not optimized along the assimilation run because we aimed our OSSE to be as less overoptimistic as possible to make sure not to over-evaluate the impact of the satellite observations, especially considering that our observations are simulated. This is why we define our choice of not evolving B matrix as a conservative one.**

**A comment was added in the revised version of the manuscript (Page 4 Lines 27-29):**

**"Following the OSSE philosophy (Timmermans et al. 2015), we want to be as less overoptimistic as possible and this is why we freeze the B matrix in time, that is to say, the same matrix is used for the whole assimilation period. The chosen correlation lengths and variance are values commonly used and tested in the MOCAGE-PALM assimilation system (e.g. Abida et al, 2017)."**

3/ Do the authors mean that the bias constitutes 10 to 18% of the tropospheric ozone level?

This needs to be mentioned in the revised version of the manuscript (page 5).

**The bias of 10 to 18% refers to the difference between LOTOS-EUROS and the Airbase database ground-based stations measurements in terms of surface ozone.**

**This comment was added in the revised version of the manuscript (Page 6 Lines 7-10):**

**"Therefore, we can conclude that LOTOS-EUROS surface ozone is consistent with the Airbase database ground-based stations measurements, both in the diurnal cycle and in the temporal evolution of the min/max pics, despite a bias ranging from 10% to 18% that could be extrapolated in the assimilated results, i.e., the assimilated results could have this bias compared to the reality and explain the low (or high) values from the assimilation."**

4/ What are the planned changes to the manuscript in response to this question of the reviewer?

Two comments were added in the revised version of the manuscript (Page 7 Lines 14-15) and (Page 7 Lines 18-21). See below:

[revised manuscript text omitted]